# Tiny Moves: Game-based Hypothesis Refinement

## Abstract

Scientific discovery is an iterative process, yet most machine learning approaches treat it as an end-to-end prediction task, limiting interpretability and alignment with scientific reasoning workflows. We introduce The Hypothesis Game, a symbolic, game-based framework where a system of agents refines hypotheses through a fixed set of reasoning moves (a reasoning grammar). Inspired by the idea that scientific progress often relies on small, incremental changes, our framework emphasizes "tiny moves" as the building blocks of incremental hypothesis evolution. We evaluate the approach on pathway-level reasoning tasks derived from Reactome, focusing on reconstruction from partial cues and recovery of corrupted hypotheses. Across 820 reconstruction and 2880 corruption experiments, it matches strong prompting baselines on reconstruction and achieves superior precision and error recovery in corruption. Beyond accuracy, it produces concise, interpretable hypotheses and enables controllable reasoning, highlighting the potential of game-based reasoning for accelerating discovery across the sciences.

## 1 Introduction

Scientific discovery is rarely a single leap from the data to the conclusion. In fields like biology, the discovery process unfolds iteratively and non-linearly. It often starts from partial hypotheses based on incomplete data, which researchers expand by combining or generating new evidence, allowing a hypothesis to evolve. The emerging hypothesis undergoes multiple rounds of pruning, testing and iterative refinement to reveal a final causal foundation (Alkan et al., 2025).

Recent work in AI for science has shown increasing interest in agentic approaches, where Large Language Models (LLMs) or multi-agent systems get assigned specialized roles, such as literature reviewer, clinical trial designer, or experiment planner, to support parts of the scientific workflow (Gridach et al., 2025; Zheng et al., 2025). Examples such as the "Co-Scientist" (Gottweis et al., 2025) and "Robin" (Ghareeb et al., 2025), as well as lab-in-the-loop multi-agent frameworks (Swanson et al., 2024) and domain-focused agent systems for biomedical discovery (Gao et al., 2024), demonstrate how role-specific capabilities and tools can be orchestrated to address domain problems end-to-end.

Although these systems integrate domain knowledge into agents' abilities, they typically leave the structure of reasoning implicit: agents produce output in free form, without clear constraints on intermediate states or transformations (Liu et al., 2023; Majumder et al., 2024). This limits interpretability, makes it difficult to control reasoning style, and hinders transfer across related problems (Mondorf & Plank, 2024; Madaan et al., 2023).

In contrast, human scientific reasoning is compositional: hypotheses are built gradually from smaller fragments and the process is guided by a repertoire of common reasoning patterns (e.g. combination, analogy, critique, generalization, expansion, etc.) (Lawson, 2004). Based on this observation we propose a symbolic, game-based framing for hypothesis refinement tasks, in which LLM agents operate over a shared hypothesis state using a fixed reasoning grammar. This grammar defines a small, generic set of moves that can be reused across a range of related biological reasoning tasks. This framing enables the system to "think about thinking" rather than hard wiring problem-specific behaviors. This grammar could in principle be applied to a variety of open-ended biological problems, from mechanism of action (MoA) construction for therapeutic drug targets to more general causal and mechanistic reasoning over complex biological processes.

In this paper, we introduce **The Hypothesis Game**, a symbolic, game-based framework for hypothesis refinement. Our contributions are threefold: (1) a formalization of hypothesis refinement as a compositional reasoning game with a reusable grammar of moves; (2) an implementation with LLM agents operating over shared hypothesis states, enabling transparent reasoning trajectories and controllable reasoning styles; and (3) an empirical evaluation on pathway-level reasoning tasks demonstrating performance competitive with strong prompting baselines, while producing finer-grained, more precise hypotheses. Together, these results highlight the potential of game-based reasoning formalisms to support more granular, interpretable, and transferable scientific discovery.

## 2 FRAMEWORK

The Hypothesis Game formalizes hypothesis refinement as the iterative transformation of a shared state through structured reasoning moves. This section defines how hypotheses are represented, how moves operate on them, and how modes and scoring functions may shape the dynamics of the game.

Here we introduce a general operator-based formalism that captures a broad design space for hypothesis refinement. We intentionally instantiate only the minimal subset of this formalism required to evaluate the central research question: whether our proposed framework with a small, reusable reasoning grammar provides measurable benefits in biological refinement tasks. Other components, such as explicit scoring, policy-based or learned controllers, and richer hypothesis representations are optional extensions of the basic game. Our experiments are designed to isolate and test the general reasoning framework, while the broader formalism outlines how more sophisticated controllers and utilities can be incorporated in future work.

### 2.1 HYPOTHESIS REPRESENTATION

A hypothesis is represented as a set of fragments:

$$H_t = \{h_1, h_2, \ldots, h_n\},$$

where each fragment $h_i$ may be a text claim, a structured triple (subject–relation–object), or optionally mapped to a graph $G = (V, E)$ of entities and relations. In our experiments, we primarily use structured text.

### 2.2 REASONING GRAMMAR (MOVES)

Let $O = \{o_1, o_2, \ldots, o_m\}$ denote a fixed set of reasoning operations. Formally, let $\mathcal{H}$ be the space of all possible hypotheses and $\mathcal{C}$ the space of contexts (e.g., cell type, disease, etc). Each operation is a function

$$o_j : \mathcal{H} \times \mathcal{C} \mapsto \mathcal{H}, \quad (H_t, C) \mapsto H_{t+1},$$

where $H_t \in \mathcal{H}$ is the current hypothesis, $C \in \mathcal{C}$ is an optional context (e.g., biological priors), and $H_{t+1} \in \mathcal{H}$ is the updated hypothesis state.

In our implementation, we restrict the set of moves to four core operations: `prune`, `expand`, `retrieve`, and `debate` (see Table 1). Moves may be atomic (e.g. `prune`, `expand`) or composite (e.g. `retrieve_expand`). More granular move types can be introduced as needed, typically informed by the structure of the underlying hypothesis representation. An example of a complete reasoning grammar based on graph representation of hypothesis fragments is shown in Fig. 1.

Moves can be applied repeatedly and composed arbitrarily. We can define a maximum number of reasoning operations per round (move budget) as a fixed constant $k_{\max}$. A round can be defined locally as one update step from $H_t$ to $H_{t+1}$, and globally, a sequence of rounds constitutes a complete game.

$$H_{t+1} = o_{j_k} \circ \cdots \circ o_{j_1}(H_t, C), \quad k \leq k_{\max}.$$

At each round, a controller selects and applies up to $k_{\max}$ moves to evolve the hypothesis. The controller can be realized in different ways (e.g., an LLM, finite state machine, or RL agent), depending on the desired game design.

## 2.3 GAME MODES

In open-ended discovery, the precise outcome is often unknown, but the overall style of reasoning can still be guided. We capture this through a *mode $M$*, which specifies how moves are selected. One way to formalise this idea is through a probability distribution over moves,

$$\pi_M(o_i \mid H_t) = P(\text{apply } o_i \mid M),$$

where, for example, a *discovery* mode favors generative moves such as `expand`, while a *validation* mode favors critical moves such as `prune` or `debate`. More generally, modes can also be realized by restricting the available moves $O$, enforcing deterministic rules, or combining weighting and constraints set by the overall objective of a game.

In our experiments, modes are approximated through natural language instructions to the controller, but the reasoning grammar provides a principled way to configure high-level exploration or validation goals in more open-ended settings.

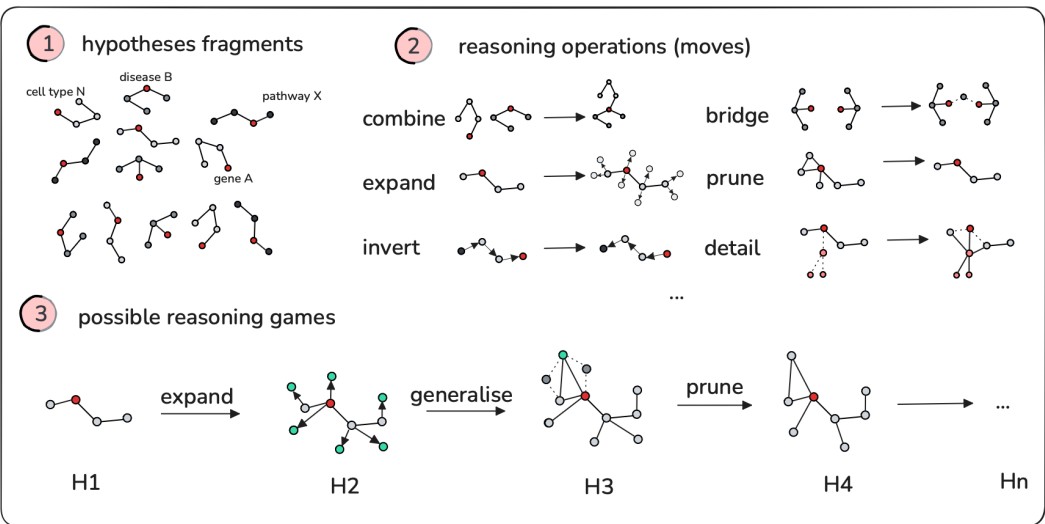

Figure 1: A conceptual framework for reasoning games. The objective of the game is to evolve a hypothesis fragment through a sequence of reasoning moves, with progress assessed through properties such as novelty, coherence, and traceability. *Graph structures shown for conceptual illustration only; actual implementation uses structured text fragments with equivalent reasoning operations.*

## 2.4 SCORING

While modes can guide reasoning styles at a high level, scoring functions may offer a way to make the game more controllable. Quantifying metrics during refinement provides a way to shape the trajectory of the game. Formally, we can define a vector of metrics,

$$S(H_t) = \left(D_{\text{known}}(H_t), \ \Delta_{\text{div}}(H_t), \ L_{\text{connect}}(H_t), \ T_{\text{frag}}(H_t)\right),$$

where the components capture distance from known hypotheses ($D_{\text{known}}$), diversity of current hypothesis ($\Delta_{\text{div}}$), local connectivity ($L_{\text{connect}}$), and traceability to prior knowledge ($T_{\text{frag}}$). These can be aggregated into a scalar utility,

$$U(H_t) = \beta^\top S(H_t),$$

with weights $\beta$ reflecting mode-specific priorities (e.g., traceability in *validation*, diversity in *discovery*). In practice, robust scoring for biological hypotheses likely requires a *hybrid* setup that combines computational metrics with sparse experimental signals, even if sparse. For example, hypothesis fragments involving molecular interactions could be evaluated using targeted binding assays or perturbation readouts, providing grounded feedback that complements algorithmic metrics.

In this work, we do not use explicit scoring to drive the controller; modes are implemented through natural-language instructions. The scoring framework presented here is therefore conceptual, illustrating how computationally and experimentally informed metrics could be integrated into more autonomous implementations in the future.

## 2.5 GAME VARIANTS

The outlined game formalism allows us to define game variants that operate on different granularity levels. **Simple Hypothesis Refinement** treats the whole hypothesis as a single state (Algorithm 1). In each round, a mode-conditioned controller selects a move from the shared grammar and updates the entire state, stopping when task goals are met.

---

**Algorithm 1** Simple Hypothesis Refinement (single round)

---

**Require:** initial hypothesis state $H_0$, reasoning moves $\mathcal{O}$, mode $M$, move budget $k_{\max}$, termination criteria
   $t \leftarrow 0$
   **while** not Terminate($H_t$) **do**
      **Game Master:** provide current state $H_t$ and mode $M$ to controller
      **Controller:** select sequence of moves $(o_{j_1}, \ldots, o_{j_k})$ with $k \leq k_{\max}$ according to $\pi_M$
      **for** each $o_j$ in selected moves **do**
         $H_t \leftarrow o_j(H_t, C)$                  ▷ apply reasoning move with optional context $C$
      **end for**
      $t \leftarrow t + 1$
   **end while**
   **return** final hypothesis $H_t$

---

Noting that large changes are rarely necessary to refine a hypothesis, we can build on the simple variant by enabling granular edits during the hypothesis' evolution. **Localized Hypothesis Refinement** keeps the same controller and move set but operates on fragments (structured text or subgraphs), selecting regions to edit and enforcing global consistency so untouched parts remain unchanged (Algorithm 2). This game type strongly depends on the underlying hypothesis representation structure.

---

**Algorithm 2** Localized Hypothesis Refinement (single round)

---

**Require:** Hypothesis state $H_t = \{h_1, \ldots, h_n\}$ (structured text or graph), moves $\mathcal{O}$, mode $M$, move budget $k_{\max}$, context $C$, selector $\sigma$
   **Selector** $\sigma$: propose a set of candidate regions $\mathcal{R} = \{R_1, \ldots, R_m\}$ where each $R_i \subseteq$ nodes/tuples of $H_t$
   **Controller** (mode $M$): choose up to $k \leq k_{\max}$ pairs $\{(o_j, R_j)\}_{j=1}^k$ with $o_j \in \mathcal{O}$
   **for** each $(o_j, R_j)$ **do**
      $H_t \leftarrow \texttt{ApplyLocal}(H_t, o_j, R_j, C)$             ▷ local rewrite on $R_j$ only
      $H_t \leftarrow \texttt{EnforceConsistency}(H_t, R_j)$      ▷ maintain schema/typing/acyclicity/etc.
   **end for**
   **return** $H_t$

---

Together, these variants illustrate that the formalism supports both high-level, whole-state reasoning and fine-grained, region-focused reasoning under a shared utility function and mode settings. The simple variant is recovered when the selected region spans the full state. This design mirrors the varying levels of complexity observed in biological systems.

## 3 IMPLEMENTATION

To test the proposed framework, we implement a minimal version of the game as a system of specialized agents, where the reasoning process is determined by a central LLM controller, **Game Master**. The Game Master guides the reasoning process by iteratively analyzing the hypothesis state

and selecting moves based on the analysis. Move selection consists of a clear request (e.g. *"remove component A from the hypothesis"*) and which agent(s) should execute it. Table 1 summarizes the moves, their components and corresponding responsibilities.

Table 1: Key elements of The Hypothesis Game. Full prompts are provided in the Supplementary Methods (see Section 3).

| Move | Components | Description |
|---|---|---|
| **Game Master (LLM controller)** | Diagnose
Move selection | Evaluate hypothesis and recommend next actions.
Choose next move based on recommendations. |
| **Prune** | Prune | Remove component(s) from hypothesis. |
| **Expand with corpus** | Retrieve evidence
Expand | Search external corpora for evidence.
Integrate retrieved information into the hypothesis. |
| **Expand with LLM introspection** | Retrieve evidence
Expand | Gather information using LLM prior knowledge.
Integrate retrieved information into the hypothesis. |
| **Debate** | Setup
Debate topic
Conclude | Frame the debate around the requested topic.
Multiple agents argue from distinct positions.
Analyse the debate and propose a final conclusion. |

**Modes:** In our minimal prototype, modes are realized by injecting mode descriptions into the initial prompt to the Game Master (controller). This prompt influences the choice of reasoning operations without an explicit probabilistic policy module. While simplified, this approach provides a controllable approximation of $\pi_M$ and allows us to explore the impact of different modes.

**Optimisation:** Game goals and stopping conditions are specified to the Game Master (controller) through the initial prompt, and the Game Master's *Diagnose* component decides when the hypothesis satisfies the requirements. Although this approach lacks explicit metric-based control, it provides a flexible mechanism for steering the game. The scoring function described above is presented as part of the general formalism, illustrating how automated, quantitative evaluation could be incorporated in future implementations.

## 4 EXPERIMENT SET-UP

Reasoning benchmarks in mathematics and common sense (GSM8K (Cobbe et al., 2021), MATH (Hendrycks et al., 2021), BIG-Bench (Srivastava et al., 2022)) do not translate to biological hypothesis generation, where researchers must build complex hypotheses step by step from incomplete, noisy, sometimes contradictory evidence rather than retrieve facts. Without established ways to evaluate reasoning quality, benchmarks should challenge systems to tolerate noise, recover missing links, and extend hypotheses in controlled ways. Emerging biological benchmarks such as BioMaze (Zhao et al., 2025) move in this direction with graph-based pathway QA and high-level LLM-as-judge evaluations, but still differ from the longer-horizon, statement-level refinement studied here.

To fill this gap, we introduce two evaluation tasks designed as first benchmarks for hypothesis refinement. These tasks mirror realistic challenges in biological discovery: (1) hypothesis reconstruction, and (2) corruption recovery (Table 2).

Table 2: Evaluation tasks overview

| Task | Purpose | Validates | Metrics |
|---|---|---|---|
| Reconstruction | Can the system rebuild known mechanisms from partial cues? | Incremental reasoning; Traceability | Precision, recall, F1 |
| Corruption Recovery | Can the system correct noisy or misleading hypotheses? | Robustness to noise; Logical refinement | Error removal rate, precision, recall, F1 |

## 4.1 TASK SETUP

We instantiate evaluation tasks using curated subsets of human pathways from Reactome (Jassal et al., 2020). Each pathway consists of biochemical reactions, available in both graph and text representations (see 1.1). In the text representation, pathways are expressed as sets of statements describing biochemical reactions; for example, *ATP phosphorylates glucose to form glucose-6-phosphate.*

We sampled pathways stratified by the number of biochemical reactions, to capture the diversity and complexity of the complete dataset. For reconstruction and corruption tasks, we sampled 100 and 20 pathways, respectively. The rationale was to create datasets large enough to capture key reasoning patterns across multiple approaches, while remaining feasible for large-scale experimentation. In total, we ran 820 experiments for reconstruction and 2880 experiments for corruption.

**Common Experimental Principles** Across all tasks, hypotheses are represented as text fragments. The Hypothesis Game is restricted to four available moves: `prune`, `expand`, `expand_with_corpus`, and `debate` (See Table 1). Move selection and termination are dynamically governed by the Game Master, adapting to task-specific goals.

We compared our approach against three reasoning baselines: Zero-Shot prompting, Chain-of-Thought, and ReAct. Zero-Shot directly generates answers without intermediate reasoning steps (Brown et al., 2020). Chain-of-Thought elicits step-by-step reasoning through intermediate natural language explanations (Wei et al., 2022). ReAct interleaves reasoning traces with access tools to improve decision making (Yao et al., 2023). We compared these baselines against our Hypothesis Game under different move configurations and a fixed move budget. All models received the same input prompt (see Supplementary A Sec. 3), which instructs the system to either reconstruct a pathway or recover a corrupted pathway. All curated datasets are available on Hugging Face[1].

**Task 1 – Reconstruction:** The reconstruction task evaluates whether a system can reconstruct complex hypotheses from partial cues by performing incremental reasoning. Starting from a minimal cue, the system must recover the biochemical reactions (steps) of a biological pathway, modeling the onerous curation process domain experts go through to construct the Reactome database. To reduce the risk of models exploiting memorized knowledge of well-known pathways, we rephrased pathway names while preserving their semantic content and level of granularity. A domain expert inspected and corrected the paraphrased titles to ensure semantic fidelity (available on Hugging Face.) For agents with tool access (our approach and ReAct), we additionally provided a corpus of open-access biomedical articles, consisting mainly of abstracts cited in the Reactome pathway descriptions.

**Evaluation** relied on two complementary notions of correctness. At the pathway level, we annotated entities (genes, protein complexes/families, and chemicals) in both original and generated pathways using Gilda (Gyori et al., 2022); precision and recall over these entity sets provided a quantitative measure of biological fidelity. At the reaction level we refer to the LLM-as-judge metric as 'Detailed Recall', it evaluates whether the generated pathways reproduced the intended biochemical reactions, assessing four attributes: input entities, output entities, reaction directionality, and type of biological interaction (Supplementary A Sec. 3). To assess the reliability of this LLM-as-judge, we conducted a post-hoc calibration study in which two senior domain experts independently scored a stratified sample of model outputs for both tasks (Supplementary A Sec. 3.8).

**Task 2 – Corruption:** The corruption task assesses the ability to detect and repair errors while preserving the structure of a valid pathway. Starting from 20 human pathways, we introduced three types of corruptions (errors) (Supplementary A Table 1):

- wrong entity – replacing a correct entity with an incorrect one;
- wrong relationship – altering the relation between entities;
- irrelevant statement – inserting a non-relevant statement into the pathway.

We further varied level of challenge along two axes: 1) **difficulty:** *easy* (trivial errors) and *hard* (subtle changes, requiring a deeper biological understanding); 2) **error rate:** 10-40% of pathway length (measured as a number of steps/reactions) to capture differences in pathway size and complexity. All

---

[1]`https://huggingface.co/datasets/TuringRRX/TinyMoves`

errors were generated by an LLM and iteratively refined, with two domain experts reviewing and manually correcting outputs to produce the curated corruption set.

**Evaluation** combined two measures. First, an LLM judge was presented with the original statement, the corrupted version, and the model's output, and determined whether the error persisted. Second, entity mapping, as in reconstruction, quantified biological fidelity by measuring precision and recall of annotated entities against the ground truth.

## 5 RESULTS

We evaluated The Hypothesis Game on two pathway-level reasoning tasks described above: reconstruction from partial cues and recovery from corrupted hypotheses. In both settings, we compare the *Hypothesis Game* configuration (four move types with access to the corpus) against strong prompting baselines (Zero-Shot, Chain-of-Thought, ReAct). This study focuses on the minimal game version, though the formalism extends to richer move sets and modes.

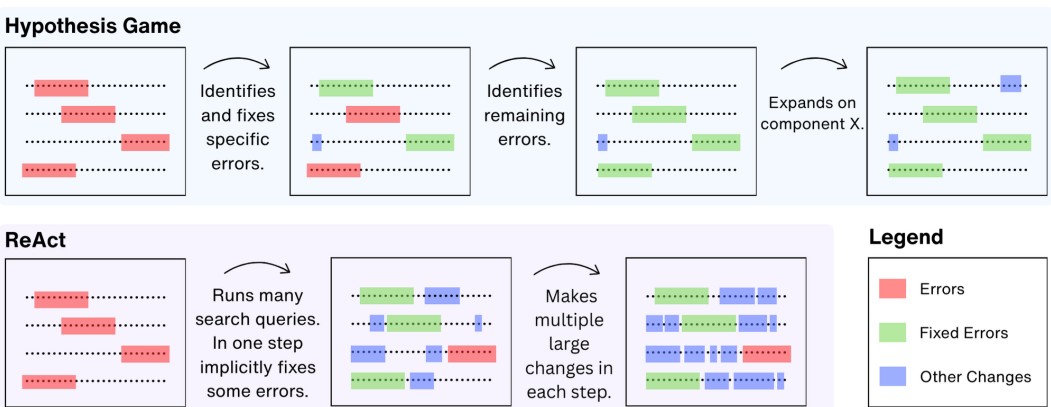

Figure 2: Representative example run of *Hypothesis Game* and ReAct on the corruption task, illustrating incremental vs large single-step edits. *\*Other changes* are quantified as (1) the number of biological entity additions/removals and (2) word-level normalised Levenshtein distance to the reference pathway. See Supplementary B Fig. 4 for details.

**Qualitative observations.** In the Reconstruction task, The Hypothesis Game tends to make smaller incremental and traceable updates to a hypothesis. In contrast, the baselines introduce larger changes at once, often overwriting significant parts of the initial hypothesis (for a complete example, see Supplementary B Sec. 1.1).

Figure 2 illustrates a similar pattern in the Corruption task. The *Hypothesis Game* incrementally identifies and corrects all errors, while making only minor additional changes to the input hypothesis. ReAct, in contrast, modifies the pathway by making multiple large changes in each step, incurring overall much larger changes to the pathway. Detailed numbers showing overall changes made to the hypothesis by each method are shown in Supplementary B Fig. 4. This highlights the benefit of controlled step-by-step refinement.

**Reconstruction task.** In the controlled reconstruction setting, the *Hypothesis Game* performed comparably to the strongest baseline (ReAct) and better than Zero-Shot and Chain-of-Thought (Fig. 3). Since some Reactome pathways are relatively well known, LLMs were expected to recall key components. This is reflected in the relatively higher recall of Chain-of-Thought and Zero-Shot. However, these methods also tended to generate hypotheses with a large number of additional concepts absent from the original pathway, leading to much lower precision, Supplementary B Fig. 1.

Overall, ReAct achieved slightly higher F1 scores than the *Hypothesis Game*, followed by Zero-Shot and Chain-of-Thought. Low precision–recall values across all methods indicate the difficulty of the pathway reconstruction task. Beyond the inherent difficulty of a task typically performed by domain experts, low performance likely reflects three factors: insufficient information in partial cues,

heterogeneity in pathway curation, and limited biological detail in an abstract-biased corpus. To better understand which reasoning moves drive performance, we performed an ablation study over all subsets of the four core moves in the reconstruction task, as well as removing access to the corpus (20 pathways; Supplementary B Table 1).

**Corruption task.** In the corruption recovery task (error rates 10–40%), the *Hypothesis Game* achieves the best overall performance. Figure 3 summarises results aggregated across pathways, corruption types, and error rates. The **Errors Removed** panel shows that *Hypothesis Game* decisively outperforms the baselines by consistently removing more errors. The **Recall** and **Precision** panels highlight the trade-off: ReAct attains high **Recall** but at the expense of **Precision**, while Chain-of-Thought and Zero-Shot retain content yet introduce additional noise. In contrast, *Hypothesis Game* combines strong error removal with the highest **Precision** and **F1 Score**, selectively pruning corrupted statements while preserving the underlying pathway structure.

The error removal panel in Figure 4 reveals a consistent hierarchy in removal difficulty. *Unsupported step* errors are most easily removed, as they introduce entire statements that are readily identified as irrelevant. *Wrong-direction* corruptions are harder, since they preserve surface plausibility while inverting causal polarity. *Wrong-entity* substitutions prove most challenging: the corrupted pathways still appear fluent, but introduce subtle inconsistencies in biochemical grounding. This shows that entity-level corruptions demand deeper semantic discrimination. Notably, *Hypothesis Game* achieved the strongest overall performance across all error types, with particularly large gains on entity and relationship errors (Fig. 4). The complete results, stratified by difficulty and corruption fraction, are provided in Supplementary B Sec. 2.1.

Overall, these results show that small, targeted reasoning moves - implemented as incremental edits rather than wholesale rewrites - enable targeted error identification and correction, yielding substantially cleaner pathway repairs than standard prompting baselines. Hypothesis Game combines strong error removal, high precision, and competitive recall across corruption types and difficulty levels, establishing it as the most effective strategy for recovering corrupted mechanistic pathways.

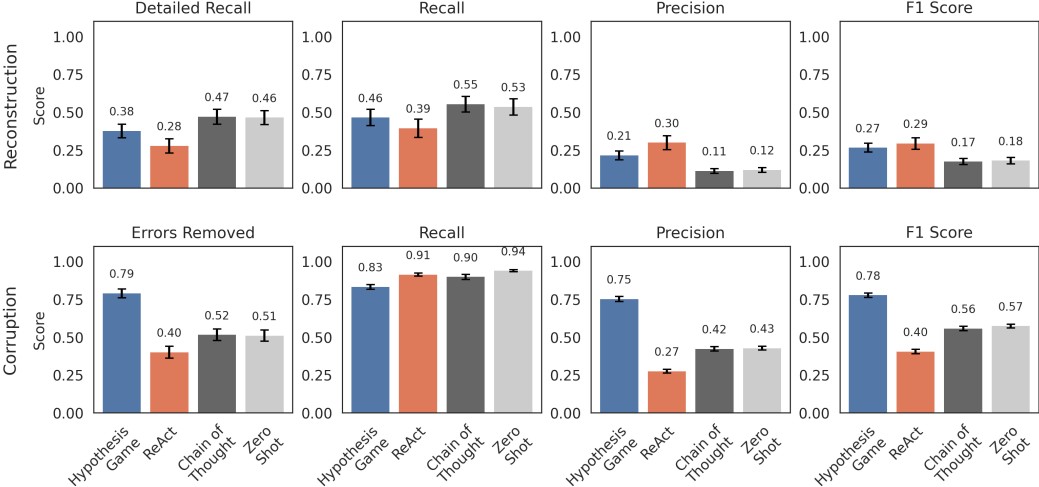

Figure 3: Comparison of *Hypothesis Game* vs. prompting baselines on two pathway-level tasks. Bars show averages over the evaluation sets described in the text. The error bars show 95% confidence intervals. Top row: **Reconstruction**; All methods struggled with faithfully reconstructing the pathways. *ReAct* and *Hypothesis Game* had a statistically non-significant difference in F1 score, but *Hypothesis Game* performed significantly better in Detailed Recall of pathways (Friedman test, $\chi^2(3) = 84.3, p < 0.0001$, post-hoc Wilcoxon test with Bonferroni correction $p < 0.001$). Bottom row: **Corruption**; *Hypothesis Game* balances error removal and retention of valid content, achieving the highest precision, F1 and error removal rate (for all scores Friedman test $p < 0.0001$, post-hoc Wilcoxon test with Bonferroni correction $p < 0.0005$).

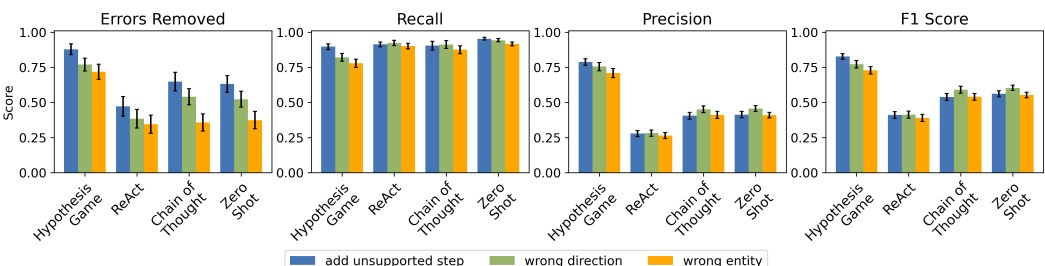

Figure 4: Aggregation of all results on the corruption task based on error type. Error bars show 95% confidence intervals.

**Summary**   Our results highlight complementary strengths across the two tasks. In reconstruction, all methods struggled, reflecting the inherent difficulty of recovering complete pathways from sparse cues. Here, the *Hypothesis Game* matched the strongest baseline (ReAct), while outperforming simpler prompting strategies in precision. In corruption recovery, the advantages of structured reasoning are evident: *Hypothesis Game* achieved the highest overall performance, combining strong error removal with superior precision and F1 scores, while maintaining recall. Taken together, these findings suggest that the game-based framework, centered on small incremental reasoning steps ("tiny moves"), is particularly effective in settings that require targeted error correction and robustness to noisy inputs. This motivates extending the approach to open-ended refinement tasks. Preliminary Monte Carlo tree search-based experiments suggest that even in such settings, the framework can generate qualitatively plausible hypotheses (Supplementary B Sec. 3), although systematic evaluation and broader experimentation are needed.

## 6   CONCLUSIONS AND FUTURE WORK

Our study demonstrates that a structured, game-based approach to hypothesis refinement can match strong prompting baselines in reconstruction tasks and clearly outperform them in corruption recovery, where explicit reasoning moves enable targeted error correction while preserving valid pathway content. These results highlight both the promise and the limitations of current methods: while controlled corruption recovery benefits strongly from structured reasoning, open-ended reconstruction remains a challenging setting for all approaches. Although our experiments focus on settings with known ground truth, the formalism can extend beyond consistency-bound refinement and can also support more exploratory hypothesis generation.

In future work we aim to extend this framework along several directions. First, we plan to systematically explore richer hypothesis representations, including structured and semi-structured text and graph formalism. Second, we plan to optimise move selection using metric-driven scoring and reinforcement learning. Third, we intend to broaden the evaluation suite to include open-ended hypothesis evolution. Taken together, these steps will move us from controlled settings with known ground truth toward more realistic discovery scenarios, enabling both consistency-driven refinement and more exploratory reasoning where robustness, novelty, and interpretability are critical.

### ACKNOWLEDGMENTS

We thank our colleagues and collaborators for their support and constructive feedback during this work. We also thank the Reactome team for making the pathway knowledge base openly available.

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
