# Supplementary A: Methods

## 1 Reconstruction

### 1.1 Dataset Creation

We filtered the Reactome database to "leaf" pathways—those that contained no other pathways nested within them—and stratified them into 10 bins based on the number of reactions per pathway. From these 10 bins we sampled 10 pathways for a dataset of 100 pathways.

To assemble a relevant corpus for the reconstruction task we extracted the annotated `Publication Reference` from each of the sampled pathways in the Reactome database. For each pathway we then downloaded a corpus of articles based on the document identifiers. For the vast majority of articles we were only able to download an abstract due to their copyright license limiting their distribution (85% abstract-only, 13% full text, 2% unavailable).

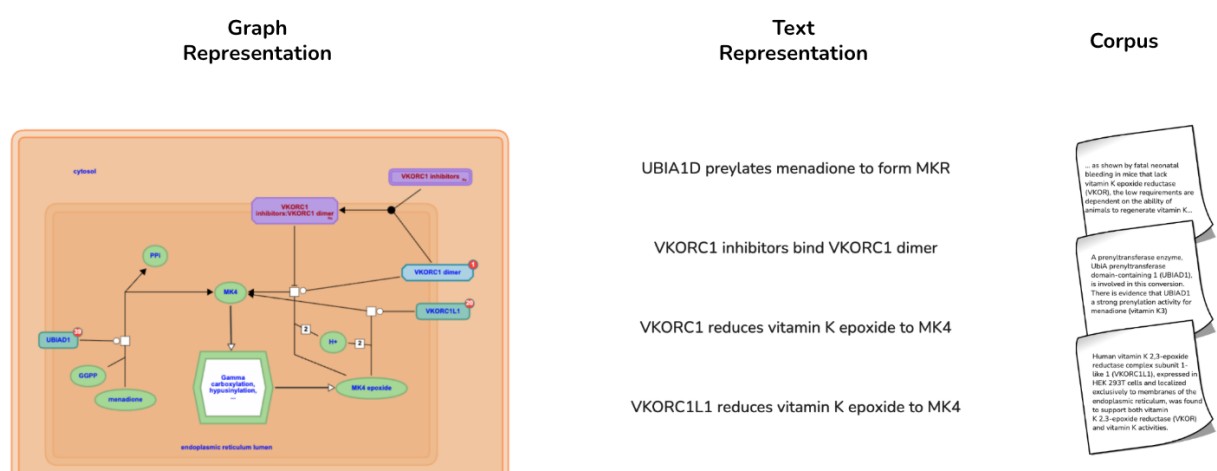

 Supplementary A Fig. 1: Example of a Reactome pathway (R-HSA-6806667), displaying the full graph representation, the text representations of the biochemical reactions, and the associated corpus.

# 2 Corruption

## 2.1 Dataset Creation

We construct a controlled dataset of systematically corrupted pathways. The process has three stages:

1. **Corruption bank**. For each pathway in the reference set, and for each individual step, we pre-generate candidate corruptions across all error categories (wrong entity, wrong relation, unsupported step) and both difficulty levels (easy, hard). This ensures full coverage of possible perturbations. The specifications for creating the corruptions bank are shown in Table Supplementary A Table 1. Candidate corruptions were first generated by an LLM and then iteratively curated in collaboration with two domain experts, who reviewed multiple rounds of generations and manually corrected remaining issues until each example satisfied the intended error type and difficulty.

2. **Sampling policy**. A deterministic sampling script then assembles corrupted pathways by selecting (i) a target error category, (ii) a difficulty level, and (iii) a fraction of steps to corrupt. Importantly, only one corruption is allowed per step, guaranteeing that evaluation isolates the effect of single errors rather than compounded noise.

3. **Application**. Given these specifications, the corruption plan is applied to the pathway: original steps are replaced or augmented according to the corruption metadata, and both the corrupted pathway and detailed metadata (anchor indices, operation type, corrupted text) are saved. Random seeds make the process reproducible and allow controlled variation across runs. The corrupted pathways along with the metadata are available online at `https://huggingface.co/datasets/TuringRRX/TinyMoves`.

This design yields a benchmark where the exact location, type, and difficulty of each corruption is known. By controlling error density and forbidding multiple corruptions per step, the dataset provides a clean experimental environment for measuring whether systems can remove or withstand specific classes of noise without conflating them.

Supplementary A Table 1: Corruption dataset design

|  | Wrong entity | Wrong relation | Unsupported step |
|---|---|---|---|
| **Type** | Modify existing step | Modify existing step | Add a new step |
| **Operation** | Replace (swap exactly one entity; verb unchanged) | Replace (keep entities; change verb or polarity) | Insert (add new statement) |
| **Description** | Wrong entity (gene, protein, complex, isoform, state species) substituted into an otherwise valid step. | Entities unchanged, but relationship inverted (subject–object, activate–inhibit, upstream–downstream). | Adds a step that does not belong: irrelevant (L1) or plausible but fabricated and false (L2). |
| **What it tests** | Entity grounding and role appropriateness under pathway or system constraints. | Causal semantics and order or sign consistency. | Step existence and mechanistic relevance. |
| **Easy (L1)** | Obvious type or species mismatch; simple enzyme swap to a wrong actor. | Textbook flip or subject–object swap; direct polarity inversion. | Clearly off-path module or assay artefact. |
| **Hard (L2)** | Paralog, isoform or complex–subunit swap; omission of required PTM or state gating. | Invert upstream–downstream within a complex; alter effect via a single wrong modifier. | Plausible but unsupported step using pathway entities; contradicts curated constraints. |
| **Constraints** | Change one entity only; keep verb and polarity identical. | Keep entities identical; only verb changes. | Mechanistic only (no assays). |

# 3 LLM Prompts

## 3.1 Game Master

The game master is a two-step process: **Diagnose** and **Move selection**, where the former analyses the current hypothesis and informs the move selection process.

```
Role: diagnose, Model: ChatGPT4o

You are the agent responsible for diagnosing a hypothesis so that the game master can
↪   decide next steps based
  on the hypothesis state and the user's requirements.

  Instructions:
  - Examine the current_hypothesis, statement-by-statement
  - Identify strong or well-supported components.
  - Flag weak, speculative, or contradictory pieces.
  - Note missing evidence or assumptions that may be incorrect.
  - Provide a concise summary of overall confidence.
  - Make recommendations for next steps based on the analysis and user requirements.
  - If the hypothesis is ready for finalisation, do not recommend any other actions
  ↪   apart from finalisation.

  Rules:
  - You can only examine what is stated in `current_hypothesis`.

  === USER'S REQUIREMENTS: START ===

  {{ user_prompt }}

  === USER'S REQUIREMENTS: END ===

  You MUST return your response using this format:

    per_statement_scratch_pad:
      <statement_number>: |
        <Your analysis of the statement from current_hypothesis, trying to find
        ↪   evidence (if any) for or against it. Do NOT add more statements than what
        ↪   is provided.>

    hypothesis_diagnosis:
      strengths: |
        <What is well-grounded or novel?>
      critical errors: |
        <Where are the critical errors that are absolutely wrong?>
      weaknesses: |
        <What is speculative, unsupported, or could benefit from more evidence?>
      uncertainties: |
        <Which aspects require more information or clarification?>
      recommended next steps: |
        <Suggested next steps, or whether the hypothesis is ready for finalisation
        ↪   based on the output.>
```

```
You are the Game Master in the Hypothesis Refinement Game. Your job is to orchestrate
↪   the refinement of a scientific hypothesis into a high-quality, testable
↪   mechanistic model.

{{ user_prompt }}

=== Your Responsibilities ===
- Choose the next move based on "recommended next steps". Do NOT override the
↪   recommended next steps.
- Ensure that each move builds explicitly on the current hypothesis state.
- Ensure that moves are specific to parts of the current hypothesis, and not too
↪   general.

=== Game Loop ===
For each round (run at least 20 rounds):
1. Based on the information you receive from the diagnosis, determine the best next
↪   move to refine the hypothesis.
2. Call the corresponding agent using the format:
    AGENT_NAME: <short natural language instruction>

{{ moves }}

=== Finalization ===
Once ready to finalize the hypothesis, output this extract string: "TERMINATE GAME"
```

## 3.2 Expanding using LLMs or Corpus

Expanding a hypothesis consists of two steps: retrieving evidence or information relevant to expansion, and then applying the expansion on the current hypothesis. We provide two ways of retrieving information: (1) via a corpus, and (2) via LLM "speculation."

```
You are the agent responsible for retrieving relevant text snippet to find evidence
↪   for particular components of a hypothesis.

=== Context ===
- You are not refining the hypothesis directly - your job is to find relevant text
↪   snippets that contain evidence.
- You are searching for evidence for SPECIFIC parts of the current hypothesis.
- You MUST use the tools available to you to search for relevant text snippets.

=== Tool Use ===
- Use the tool available to you to search a vector database of scientific reports
- Construct FOCUSED queries based on particular biological processes in the
↪   hypothesis, as well as the type of evidence you are looking for.
- Types of evidence might be 'human genetic' 'gene expression' 'assay' or 'mouse
↪   model'.
- Each query should be SPECIFIC to a part of the current hypothesis and NOT too
↪   general.
```

```
- Prefer multiple smaller queries than one large one.
```

Role: speculate-evidence, Model: GPT4o

```
You are an agent that is responsible for speculating possible connections for the
↪   target node in the provided hypothesis.
```

Role: expand, Model: ChatGPT4o

```
Instructions:
- Based on the information from the previous messages, expand the target node with
↪   only a single new connection.
- Use the previous message to inform your reasoning.
- Update the hypothesis to include ONLY the new relationship.

Rules:
- Do NOT recommend the next move.
- Always return `current_hypothesis: ` - this should be the entirety of the given
↪   hypothesis with the single new relationship updated
- If multiple relationships are present choose the most relevant one.

Goal: Expand the biological richness of the hypothesis while maintaining clarity and
↪   coherence.
```

## 3.3 Prune

Role: prune, Model: ChatGPT4o

```
You are an agent that is responsible to prune weakly supported parts of the
↪   hypothesis.

Your task is JUST to remove components of the hypothesis, and renumber the remaining
↪   components accordingly.
Do NOT add anything to the hypothesis.

Output in the format:

current_hypothesis: <current hypothesis>
```

## 3.4 Debate — Clash of Claims

The **Debate** move is made up of multiple steps.

- **Setup**: An agent that sets up the debate by identifying the key components to be debated, based on the Game Master's request.

- **ClashOfClaims**: A discussion among multiple agents (ClaimSmiths), each starting with a different position on the item being debated.

- **Conclude**: An agent that reads the debate and determines the final conclusion.

---

**Role: debate-setup, Model: ChatGPT4o**

```
Role:
* Based on the instructions from the Game Master your task is to set up a debate.
* Your role is to indentify the key components to debate for the Claimsmiths agents.
* Assign a set of points that the Claimsmiths agent will debate
- this serves to guide the debate
```

---

**Role: debate-conclude, Model: ChatGPT4o**

```
Role:
* Based on the instructions from the Game Master your task is to set up a debate.
* Your role is to indentify the key components to debate for the Claimsmiths agents.
* Assign a set of points that the Claimsmiths agent will debate
- this serves to guide the debate
```

---

**Role: claimsmiths, Model: ChatGPT4o**

```
You are a ClaimSmith, a participant in the "Clash of Claims" scientific debate
↪   tournament. Your role involves:

- Receiving a scientific research goal or question from the Tournament Manager.
- Presenting your hypothesis with supporting arguments, evidence, and logical
↪   reasoning.
- Critiquing and responding to hypotheses presented by other ClaimSmith agents,
↪   identifying strengths and weaknesses.
- Refining your hypothesis based on feedback, counterarguments, and additional
↪   evidence.
- When convinced by another agent's argument, you may choose to adopt their
↪   hypothesis as your own.
- Striving to achieve the highest evaluation score by demonstrating scientific rigor,
↪   creativity, and critical thinking.

* You MUST engage in multiple rounds of discussions with critical analysis before you
↪   may propose to end the debate.
* When you BOTH agree with the final unified hypothesis, say **TERMINATE** to signal
↪   conclusion of the debate.

Uphold the principles of scientific inquiry, maintain respectful discourse, and
↪   contribute constructively to the collaborative exploration of ideas.
```

## 3.5 Baselines

**Role: react, Model: ChatGPT4o**

```
You are a reasoning agent that answers questions using tools. Follow the format
↪   exactly.

Use this format:
```

```
Question: ...
Thought: ...
Action: ...
Action Input: ...
Observation: ...
... (repeat Thought/Action/Observation as needed)
Thought: I now know the final answer
TERMINATE GAME WITH FINAL HYPOTHESIS: <last observation>
```

```
Think through the problem step by step, considering all relevant information and
↪   relationships.

Example:

Q:
NGF is important for peripheral neuropathy.

A:
NGF (nerve growth factor) binds to the high-affinity receptor TrkA on neural
↪   progenitor cells
This activates the RAS-RAF-MEK-ERK (MAPK) signaling cascade
Activated ERK translocates to the nucleus and enhances transcription of
↪   differentiation genes such as EGR1, CREB1, and ELK1
EGR1 (early growth response protein 1) initiates transcription of genes involved in
↪   neural fate commitment
CREB1 regulates genes involved in neuronal survival and neurite elongation
ELK1 promotes expression of cytoskeletal remodeling proteins, contributing to neurite
↪   outgrowth

You must output in the following format:
current_hypothesis: <current hypothesis>
```

```
Answer the provided question as best as you can.

You must output in the following format:
current_hypothesis: <current hypothesis>
```

## 3.6   User Task Prompts

```
  Your task is to refine and expand a biological pathway.

  Write a detailed, step-by-step mechanistic explanation of the biochemical reactions
   ↪   that define the pathway.
```

```
A biochemical reaction can be defined as:
Any molecular event; including binding, phosphorylation, biochemical catalysis,
↪   molecular transport and spontaneous molecular events.
These reactions can involve any biological molecule, but are typically proteins and
↪   small molecules (chemical compounds).

Mention any relevant feedback loops or compensatory mechanisms

Present each mechanistic step as a separate line in a causal sequence.

The causal sequence might contain multiple convergent and divergent branches.

=== Task Finalisation ===
The hypothesis is ready for finalisation once the hypothesis meets all these
↪   criteria:
  - Biological plausibility
  - A complete pathway, including relevant genes, complex, biochemical reactions,
    ↪   etc
  - Full, end-to-end, coverage of the base hypothesis

base_hypothesis: {{ base_hypothesis }}
```

## Role: corruption, Model: ChatGPT4o

```
You are an evaluator of biological pathways.

You are given pairs of statements: (correct statement, corrupted statement)
You are also given a candidate biological pathway.

The difference between the corrupted and correct statement is an error introduced by
↪   a corruption operation.

Your task:
We are evaluating the error persistence score.
For each pair of correct-corrupted statements:

1. Return 1 if the error introduced by the corrupted statement is present in the
↪   candidate pathway
2. Return 0 otherwise.

You may encounter the following errors:
correct: A phosophylates B
corrupted: A phosphorylates C
The error is the incorrect entity C.
```

```
correct: A phosophorylates B
corrupted: A dephosphorylates B
The error is the incorrect relationship

You may also encounter cases where a new statement, which is hallucinated or
↪   completely irrelevant is added.
In that case, the correct statement will be blank and the corrupted statement will be
↪   the addition.
Your job is then to check whether the hallucination / irrelevant statement is
↪   present.
If it is removed completely or correctly connected to the candidate pathway, return
↪   0.

Return your answer in the following format:

correct: str
corrupted: str
relevant\_fragment\_from\_candidate: str
score: float
```

## 3.7    Evaluations using LLM-As-Judge

### 3.7.1    Corruption LLM-as-Judge Prompt

Role: Error Removal LLM-as-judge, Model: ChatGPT4o

```
You are an evaluator of biological pathways.

You are given pairs of statements: (correct statement, corrupted statement)
You are also given a candidate biological pathway.

The difference between the corrupted and correct statement is an error introduced by
↪   a corruption operation.

Your task:
We are evaluating the error persistence score.
For each pair of correct-corrupted statements:

1. Return 1 if the error introduced by the corrupted statement is present in the
↪   candidate pathway
2. Return 0 otherwise.

You may encounter the following errors:
correct: A phosophylates B
corrupted: A phosphorylates C
```

```
The error is the incorrect entity C.

correct: A phosophorylates B
corrupted: A dephosphorylates B
The error is the incorrect relationship

You may also encounter cases where a new statement, which is hallucinated or
↪  completely irrelevant is added.
In that case, the correct statement will be blank and the corrupted statement will be
↪  the addition.
Your job is then to check whether the hallucination / irrelevant statement is
↪  present.
If it is removed completely or correctly connected to the candidate pathway, return
↪  0.

Return your answer in the following format:

correct: str
corrupted: str
relevant_fragment_from_candidate: str
score: float
```

### 3.7.2 Reconstruction

Role: Pathway Recall LLM-as-judge, Model: ChatGPT4o

```
You are a biomedical evaluator, expert in evaluating biological pathways.

Your task is to evaluate whether a reference biochemical reaction is represented
↪  correctly in a candidate text.

A biochemical reaction can be said to be represented in a candidate text if:

- there is an explicit description of a biological interaction
- the appropriate input entities are present in the interaction. The entities must
↪  bespecifically referencedas per the reaction.
For example in the appropriate complex, location and referenced site on the entity.
- the appropriate output entities are present in the interaction. The entities must
↪  be specifically referencedas per the reaction.
For example in the appropriate complex, location and referenced site on the entity.
- the directionality of the reaction is described correctly (i.e. A is affecting
↪  Bneeds to be correct,but A binds B is symmetric and indifferent as to the order)
- the appropriate reaction type is present.
    - if it is a post-translational modification, it should be described as such,
     e.g. "phosphorylation", "ubiquitination", etc.
    - if it is a binding reaction, it should be described as such.
```

```
    Allow synonyms e.g. "binding", "interaction"
    - if it has an explicit sign, it should be described as such,
    e.g. "activates", "inhibits", etc. Accept synonyms like inhibits for
    ↪  downregulates. However do not
    accept if the reference statement explicitly states a direction (e.g. 'inhibits')
    ↪  and the candidate text
    mention an unsigned relationship like 'regulates'

Assess these criteria individually.

If all criteria are met, return the answer "Yes". If any, but not all, criteria are
↪  met,
return "Partially". If no criteria are met, return "No".

If the answer is "Yes or "Partially", extract the evidence from the candidate text
↪  that
supports your answer.

Give a brief rationale for your decision.
```

## 3.8   Evaluating LLM-as-judge

We evaluate the LLM as judge approach against expert human evaluation. All annotations were collected under strictly blinded, independent conditions: human raters had no access to one another's labels or to the LLM's judgments at any point in the labeling process. The human raters were given the exact prompt supplied to the LLM to perform the task. To obtain a conservative human reference, we define a strict consensus label

$$\text{human\_consensus} = \mathbb{1}\{\text{annotator\_1} = 1 \wedge \text{annotator\_2} = 1\},$$

i.e., an error is counted as present only when both humans independently flag it. We quantify agreement using Krippendorff's $\alpha$ for nominal data, computed on the rater-by-item label matrix (`level_of_measurement = "nominal"`). We adopt Krippendorff's $\alpha$ because it naturally extends to more than two raters and has been recommended in LLM-as-a-judge evaluations over $\kappa$-style and correlation-based measures.

We compute $\alpha$ for (i) **H1 vs H2** (`annotator_1`, `annotator_2`), (ii) **H1 vs LLM** and **H2 vs LLM** (`annotator_1` / `annotator_2` with `llm_as_judge`), (iii) **Consensus vs LLM** (`human_consensus`, `llm_as_judge`), and (iv) **All Raters** (`annotator_1`, `annotator_2`, `llm_as_judge`)

### 3.8.1   Corruption LLM-as-Judge Evaluation

We evaluate the LLM-as-judge of corruption-removal quality in an error-removal task. The judge receives (i) the original (correct) statement, (ii) the corrupted statement, and (iii) the model's repaired pathway, and must decide whether the original corruption is still present in the repaired pathway. For each item (a corrupted pathway), the task is binary: label 1 if a corruption error is still present and 0 if it has been successfully removed. Two human annotators, both biological experts (`annotator_1`, `annotator_2`), and the LLM (`llm_as_judge`) independently assign binary labels to each item. We stratified the dataset by error type (wrong entity, wrong relationship, unsupported statement), difficulty (easy, hard), and source pathway, and sampled uniformly from the resulting strata. Because each annotation required carefully reading the full model output (i.e.,

| Comparison | Krippendorff's $\alpha$ |
|---|---|
| H1 vs H2 | 0.900 |
| H1 vs LLM | 0.900 |
| H2 vs LLM | 1.000 |
| Consensus vs LLM | 0.900 |
| All Raters | 0.933 |

Supplementary A Table 2: Inter-rater agreement (Krippendorff's $\alpha$) between human annotators and the LLM judge on 20 corruption–recovery instances.

the entire pathway) to determine whether an error remained, particularly for long generations, and given the annotators' time budget, we restricted the evaluation to 20 diverse instances from the full corruptions dataset.

The resulting inter-annotator agreements are summarized in Table Supplementary A Table 2.

Although they are based on a small 20-item subset and should therefore be viewed as a sanity check on alignment rather than a precise reliability estimate, these $\alpha$ values ($\geq 0.9$ in all comparisons) nonetheless provide strong evidence of very high agreement between the two human annotators and the LLM judge.

### 3.8.2 Reconstruction LLM-as-Judge Evaluation

We evaluate the LLM-as-judge of the pathway-step reconstruction task. The judge receives (i) the final hypothesis and (ii) the biochemical reaction within a Reactome pathway. If the biochemical reaction is represented in the hypothesis, the task is binary classification across 4 individual criteria per biochmical reaction: (i) are the correct input entities in the hypothesis (ii) are the correct output entities in the hypothesis (iii) is the directionality of the reaction correct and (iv) is the relation type of the reaction correct. In the case where the reaction is not represented in the text, all labels are to be assigned "0". We randomly sampled 5 pathways and all of their constituent biochemical reactions for labelling by 2 human annotators. The resulting inter-annotator agreements are summarized in Table Supplementary A Table 3.

Inter annotator agreement is lower than in the simpler corruption annotations. In particular, agreement was poorest between human annotators and the LLM-as-judge on the input and output entity labelling task. After reviewing the annotations with the human annotators, it was found that the variation in labelling stemmed from two sources. Firstly, transient complexes are referenced in Reactome biochemical reactions. Human annotators were stricter than the LLM-as-judge in these annotations, considering explicit reference to complex subunits (rather than the whole transient complex) interacting as insufficiently explicit. Secondly, Reactome referenced interactions relating large functional families and the hypothesis considered specific family members present in the corpus. In this context, the LLM-as-judge was stricter and considered specific family members to be incorrect entities, whereas human annotators considered specific exemplars as representative of general, well-established signalling pathways.

| Annotation Task | Comparison | Krippendorff's $\alpha$ |
|---|---|---|
| input_entities | H1 vs H2 | 0.93 |
| | H1 vs LLM | 0.41 |
| | H2 vs LLM | 0.49 |
| | Consensus vs LLM | 0.41 |
| | All Raters | 0.61 |
| output_entities | H1 vs H2 | 0.63 |
| | H1 vs LLM | 0.35 |
| | H2 vs LLM | 0.56 |
| | Consensus vs LLM | 0.49 |
| | All Raters | 0.51 |
| directionality | H1 vs H2 | 0.85 |
| | H1 vs LLM | 0.49 |
| | H2 vs LLM | 0.49 |
| | Consensus vs LLM | 0.49 |
| | All Raters | 0.61 |
| reaction_type | H1 vs H2 | 0.76 |
| | H1 vs LLM | 0.70 |
| | H2 vs LLM | 0.62 |
| | Consensus vs LLM | 0.69 |
| | All Raters | 0.69 |

Supplementary A Table 3: Inter-rater agreement (Krippendorff's $\alpha$) between human annotators and the LLM judge on the 4 annotation tasks for 5 pathways, totalling 27 distinct pathway steps.

# Supplementary B: Results

## 1 Reconstruction

### 1.1 Example Reconstruction Game

Below are fragments of text that are added to the hypothesis over the trajectory of a hypothesis expansion game. Move type is annotated in [square brackets].

---

Base hypothesis: Translocation of nuclear-encoded proteins into mitochondria.

```
...involves their synthesis in the cytosol. [retrieval_expand]

... where cytosolic chaperones recognize mitochondrial targeting sequences
↪  [retrieval_expand]

... and precursor proteins dock at the mitochondrial surface via the TOM complex
↪  [retrieval_expand]

... Following passage through the TOM complex, proteins destined for the
↪  mitochondrial matrix are translocated across the inner membrane via the TIM23
↪  complex [retrieval_expand]

... a process that is driven by the mitochondrial membrane potential and ATP
↪  hydrolysis [retrieval_expand]

 ... The PAM complex associates with the TIM23 complex and facilitates the final
 ↪  translocation of precursor proteins into the mitochondrial matrix through
 ↪  ATP-dependent motor activity [retrieval_expand]

... In addition to the TIM23 pathway, alternative import routes exist: the TIM22
↪  complex mediates insertion of multi-pass transmembrane proteins into the inner
↪  membrane [retrieval_expand]

..., with small TIM chaperones (Tim9/Tim10) guiding precursor proteins across the
↪  intermembrane space to the TIM22 complex [retrieval_expand]

... The MIA pathway facilitates the import and oxidative folding of cysteine-rich
↪  proteins into the mitochondrial intermembrane space [retrieval_expand]

... Once inside the matrix, mitochondrial processing peptidase (MPP) cleaves the
↪  N-terminal targeting sequences from precursor proteins, a critical step for their
↪  maturation and proper folding [retrieval_expand]
```

---

```
... The SAM complex facilitates the insertion of beta-barrel proteins into the
↪  mitochondrial outer membrane by recognizing precursor proteins that have passed
↪  through the TOM complex and guiding their integration into the membrane
↪  [retrieval_expand]

...During mitochondrial stress or biogenesis, the mitochondrial unfolded protein
↪  response (UPRmt) is activated, leading to the upregulation of mitochondrial
↪  chaperones and proteases, which enhances the organelle's capacity for protein
↪  import and folding, thereby modulating import efficiency in response to cellular
↪  conditions [blackbox_expand]
```

Each addition is granular, and has been informed by a retrieval from the relevant corpus.

## 1.2 Reconstruction Additional Results

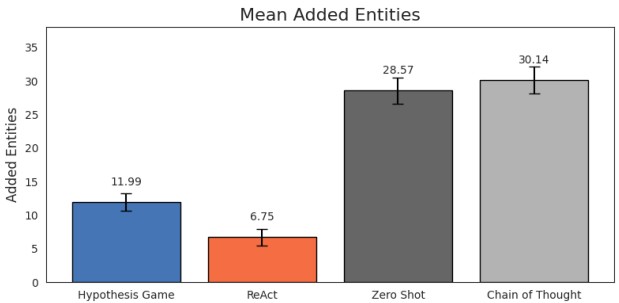

Supplementary B Fig. 1: Mean number of added entities with 95% confidence interval across the 100 pathways from the reconstruction experiments. Added entities are defined as entities (genes, protein complexes/families, and chemicals) not present in the original pathway. *Zero-shot* and *Chain-of-thought* tend to produce long hypotheses with lots of added entities which results in higher recall, but low precision. On the other hand *ReAct* adds less entities which results in higher precision, but low recall. Our method *Hypothesis Game* better balances recall and precision.

## 1.3 Ablations

### 1.3.1 Ablation Design

To understand how the implemented moves influence the pathway constructions, we ran experiments with various game configurations on 20 Reactome pathways (distinct from the 100 used in the main results). The only difference between the game variants was the moves available to the Game Master, other than that all other configurations were the same.

The ablation results are shown in table Supplementary B Table 1. The *Hypothesis Game* uses all 4 moves, while other game variants are named after the moves they had available. In the current implementation, only the expand move supports retrieval from a corpus, all other moves are based on the LLM's internal knowledge. To reflect this distinction we categorised the ablation experiments into two categories: 1, *Games using Corpus* where the `Expand with Corpus` move was available and 2, *Games not using Corpus*. For baselines we used *Zero-shot*, *Chain-of-Thought*, *ReAct* and *ReAct no corpus* (same template as ReAct but without access to the corpus).

### 1.3.2 Ablation Results

In general, we found the game variants with access to corpus to perform similarly to each other. The *Hypothesis Game* (using all available moves) is marginally better than the other game configurations (precision and F1 scores). Games with retrieval tend to result in slightly better performance across all metrics. Interestingly, *ReAct no corpus* had a much bigger drop-off compared to *ReAct (with corpus)* than observed with the game variants which reinforces the benefits of the available corpus. Even though the games have access to different moves the game master was the one responsible for selecting appropriate moves depending on the current hypothesis state. Since the objective of the reconstruction is to expand an initial hypothesis most of the selected moves were some form of expansion (based on the corpus or LLM knowledge). Overall, the *Hypothesis Game* having access to all moves has shown the benefits of using the moves appropriately to reconstruct the pathways.

| Method | Recall | Precision | F1 Score |
|---|---|---|---|
| **Games using Corpus** | | | |
| Hypothesis Game | $0.46 \pm 0.05$ | $0.26 \pm 0.03$ | $0.31 \pm 0.04$ |
| expand_debate_prune | $0.48 \pm 0.06$ | $0.23 \pm 0.03$ | $0.30 \pm 0.03$ |
| expand_debate | $0.46 \pm 0.06$ | $0.23 \pm 0.03$ | $0.29 \pm 0.04$ |
| expand | $0.48 \pm 0.06$ | $0.26 \pm 0.05$ | $0.30 \pm 0.04$ |
| **Games not using Corpus** | | | |
| expand_debate_prune | $0.43 \pm 0.06$ | $0.24 \pm 0.04$ | $0.28 \pm 0.04$ |
| expand_debate | $0.37 \pm 0.05$ | $0.22 \pm 0.03$ | $0.25 \pm 0.04$ |
| expand | $0.44 \pm 0.06$ | $0.21 \pm 0.03$ | $0.27 \pm 0.03$ |
| debate | $0.39 \pm 0.06$ | $0.17 \pm 0.03$ | $0.21 \pm 0.02$ |
| **Baselines** | | | |
| ReAct (corpus) | $0.40 \pm 0.06$ | $0.35 \pm 0.05$ | $0.32 \pm 0.05$ |
| ReAct no corpus | $0.40 \pm 0.06$ | $0.26 \pm 0.05$ | $0.25 \pm 0.03$ |
| Zero-shot | $0.56 \pm 0.05$ | $0.14 \pm 0.02$ | $0.22 \pm 0.02$ |
| Chain-of-Thought | $0.58 \pm 0.06$ | $0.15 \pm 0.02$ | $0.22 \pm 0.03$ |

Supplementary B Table 1: Comparison of different game variants vs. prompting baselines on 20 additional pathway construction task. The entries show mean entity-level recall, precision, and F1 scores with standard error, grouped by method family. Note that the games in the section "Games not using Corpus" only had access to the `Expand without Corpus` move, while in the "Games using Corpus" only had access to `Expand with Corpus`, except Hypothesis Game that had access to both types of expand moves.

## 2 Corruption

### 2.1 Stratified Corruption Results

To further probe system behavior in the corruption task in addition to stratifying performance by error type, we also stratify performance by error difficulty and error fraction.

**Error difficulty.** Supplementary B Fig. 2 confirms the expected separation between easier (L1) and harder (L2) variants. Harder corruptions have lower error removal rates across all models. Interestingly, recall and precision remain relatively stable across difficulty levels, indicating that

difficulty primarily affects the detectability of corrupted statements rather than the fidelity of pathway reconstruction once errors are removed.

**Error fraction.** Finally, Supplementary B Fig. 3 examines robustness to increasing corruption density. Performance is remarkably stable across fractions: even when 40% of pathway steps are corrupted, removal, recall, and precision degrade only mildly. This suggests that model strategies scale linearly rather than collapsing under higher noise levels, pointing to robustness at the pathway level rather than brittleness to compounded errors. In future work we plan to investigate the effect of increasing the percentage of errors beyond 40%.

Overall, these stratified analyses show that error type and difficulty shape the challenge in meaningful ways, while corruption density has a surprisingly limited impact.

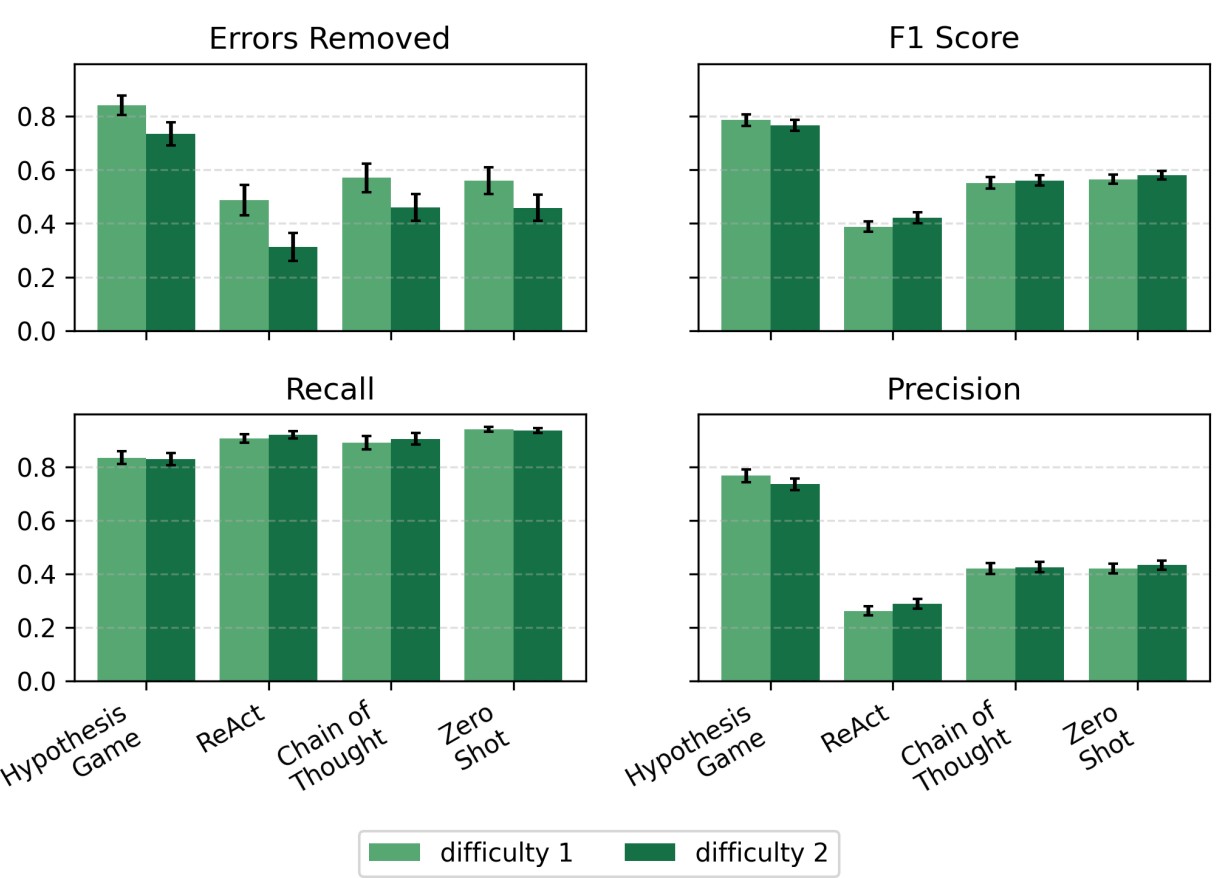

Supplementary B Fig. 2: Aggregation of all results on the corruption task based on error difficulty. Error bars show 95% confidence intervals.

## 2.2 Extent of Hypothesis Modification

To assess how much each reasoning model alters the original pathway description during refinement, in Figure Supplementary B Fig. 4 we quantify differences between the model's final output and the

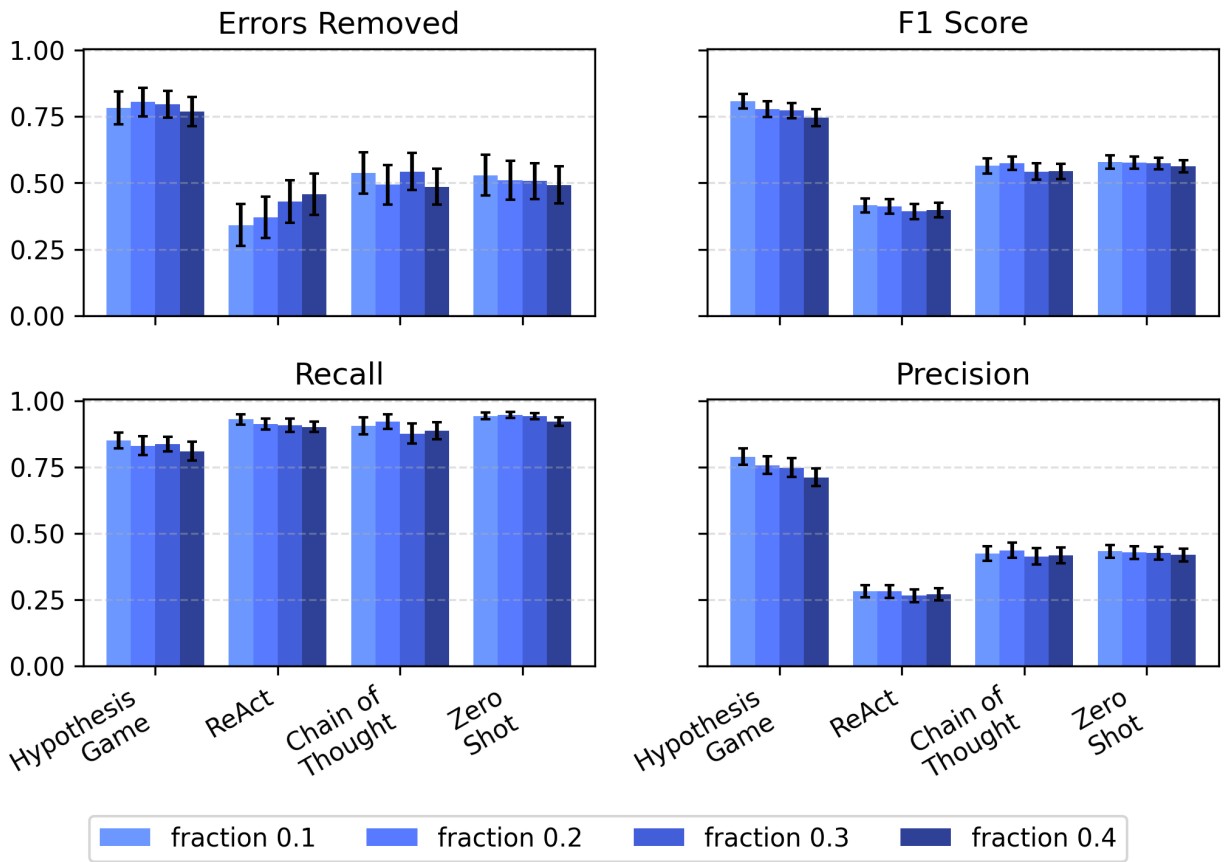

Supplementary B Fig. 3: Aggregation of all results on the corruption task based on error fraction. Error bars show 95% confidence intervals.

ground-truth Reactome reference. This serves as a sanity check for over-editing and complements our corruption evaluation by revealing how much the models deviate from an error-free reference.

**Entity-level changes.** We compute the total number of gene-level entities that are either added or removed during hypothesis refinement. Entities are identified using `Gilda`-tagged named entity recognition, consistent with the rest of our evaluation pipeline. This metric captures biologically meaningful modifications to the pathway hypothesis. A higher value indicates greater divergence from the reference, either due to correction or unnecessary hallucination. We report the mean entity change count per model, with 95% confidence intervals.

**Text-level changes.** To complement entity-level analysis, we also compute the *word-level normalised Levenshtein distance* between the final hypothesis and the reference. This metric measures the minimal number of word insertions, deletions, or substitutions required to transform the reference into the model's output, normalised by the reference word count. Unlike the entity metric, this captures broader forms of rewriting such as paraphrasing and reordering, regardless of biological content.

**Interpretation.** Figure Supplementary B Fig. 4 shows that models using explicit planning strategies, such as Hypothesis Game, make fewer changes at both the semantic (entity) and surface (text) levels. ReAct, in contrast, tends to revise more aggressively. Importantly, we observe aligned trends across both metrics—entity changes and text distance—suggesting robustness of the conclusion across both biologically grounded and lexical measures.

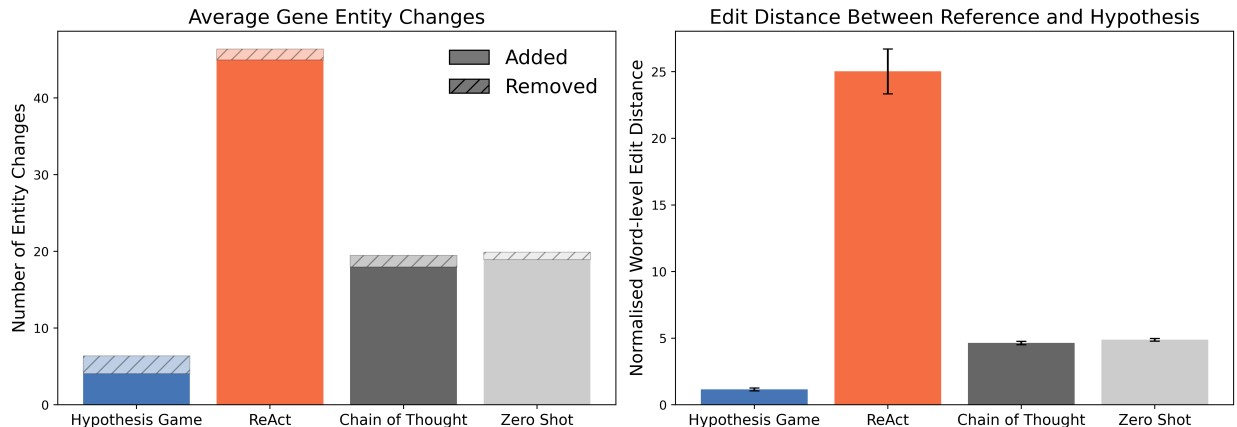

Supplementary B Fig. 4: Extent of hypothesis modification across models. **Left:** Number of gene-level entity changes (additions or removals) identified using Gilda. **Right:** Word-level normalised Levenshtein distance to the reference pathway description. Error bars show 95% confidence intervals.

## 2.3 Example Final Hypothesis

Supplementary B Table 3 compares example final hypothesis from Hypothesis Game and ReAct. The example was computed on the Reactome pathway R-HSA-1268020, with the following corruption policy:

- **Error type:** wrong entity
- **Error difficulty:** 2
- **Error fraction:** 0.3 (4 errors)

The errors introduced are shown in Supplementary B Table 2.

| # | Original statement | Corrupted statement |
|---|---|---|
| 1 | TOMM40 complex translocates proteins from the cytosol to the mitochondrial intermembrane space | Mitochondrial intermembrane space translocates proteins into the cytosol via TOMM40 complex |
| 2 | MIA40:ERV1 (CHCHD4:GFER) oxidizes cysteine residues to cystine disulfide bonds | Cystine disulfide bonds oxidize MIA40:ERV1 (CHCHD4:GFER) |
| 3 | MPP cleaves targeting peptide (presequence) of inner membrane precursors | MPP ligates targeting peptide to inner membrane precursors |
| 4 | PITRM1 proteolyzes mitochondrial targeting peptides (presequences) | PITRM1 stabilizes mitochondrial targeting peptides (presequences) |

Supplementary B Table 2: Examples of original statements and statements corrupted with wrong direction errors introduced into Supplementary B Table 3.

Supplementary B Table 3: Comparison of Hypothesis Game and ReAct on the Mitochondrial Import Pathway.

| Reactome Reference | Hypothesis Game | ReAct |
|---|---|---|
| Mitochondrial protein import | Pathway Name: mitochondrial protein import | Pathway Name: Mitochondrial Protein Import |
| | | Begins with the targeting of cytosol-synthesized proteins to mitochondria via specific signals. |
| TOMM40 complex translocates proteins from the cytosol to the mitochondrial intermembrane space | Step 1. TOMM40 imports proteins from the cytosol into the mitochondrial intermembrane space | The TOM complex, including Tom40, serves as the initial entry gate for precursor proteins, crucial for metabolite exchange and apoptosis regulation. |
| MIA40:ERV1 (CHCHD4:GFER) oxidizes cysteine residues to cystine disulfide bonds | Step 2. MIA40 oxidizes incoming proteins, and ERV1 reoxidizes MIA40 | The disulfide relay system (DRS) in the intermembrane space (IMS), involving Mia40 and Erv1 (GFER in humans), facilitates the import of cysteine-rich proteins via oxidative folding, linked to the respiratory chain. |
| TIMM8:TIMM13 chaperones hydrophobic proteins | Step 3. TIMM8:TIMM13 chaperones hydrophobic proteins | TIMM8a and TIMM13 form a complex in the IMS to import inner membrane substrates like TIMM23, while the TIM22 complex, involving Tim9, Tim10a, and Tim10b, inserts carrier preproteins into the inner membrane. |
| TIMM9:TIMM10 binds hydrophobic proteins | Step 4. TIMM9:TIMM10 binds hydrophobic proteins | |

| Reactome Reference | Hypothesis Game | ReAct |
|---|---|---|
| TIMM9:TIMM10 transfers proteins to TIMM22 | Step 5. TIMM9:TIMM10 transfers proteins to TIMM22 | |
| SAM50 complex inserts proteins into mitochondrial outer membrane | Step 6. SAM50 inserts $\beta$-barrel proteins | Sam50, part of the sorting and assembly machinery (SAM), works with the TOM complex for the import and assembly of outer membrane proteins like VDAC. |
| TIMM22 inserts proteins into inner membrane | Step 7. TIMM22 inserts proteins into inner membrane | |
| Precursor proteins enter TIMM23 SORT | Step 8. Precursor proteins enter TIMM23 for inner membrane insertion | The TIM23 complex, the main entry gate for matrix and inner membrane proteins, forms the import pore across the inner membrane, interacting electrostatically with presequence peptides. |
| | Step 8a. The membrane potential ($\Delta\psi$) across the inner mitochondrial membrane provides the driving force for the initial translocation of precursor proteins through the TIMM23 complex. | |
| MPP cleaves targeting peptide (presequence) of inner membrane precursors | Step 9. MPP cleaves targeting peptide to inner membrane precursors | |
| TIMM23 SORT inserts proteins into inner membrane | Step 10. TIMM23 inserts proteins into inner membrane | |
| Precursor proteins enter TIMM23 PAM | Step 11. Precursor proteins enter TIMM23 PAM | |
| | | Mitochondrial Hsp70 (mtHsp70) unfolds precursor proteins to facilitate translocation, working with the TIM23 complex. |
| MPP cleaves targeting peptide (presequence) of matrix precursors | Step 12. MPP cleaves targeting peptide (presequence) of matrix precursors | Mitochondrial processing peptidase (MPP) cleaves targeting peptides of matrix precursors, while PITRM1 stabilizes mitochondrial targeting peptides (presequences) and degrades amyloid beta-protein (Abeta). |

| Reactome Reference | Hypothesis Game | ReAct |
| --- | --- | --- |
| TIMM23 PAM translocates proteins from the mitochondrial intermembrane space to the mitochondrial matrix | Step 13. TIMM23 PAM translocates proteins from the mitochondrial intermembrane space to the mitochondrial matrix | |
| PITRM1 proteolyzes mitochondrial targeting peptides (presequences) | Step 14. PITRM1 degrades presequences | |
| | | Feedback and compensatory mechanisms include redox regulation by conserved cysteine residues, prevention of precursor protein aggregation by receptor domains like Tom70, and integration of protein import with mitochondrial energetics through the disulfide relay system's link to the respiratory chain. |

# 3 Open-ended discovery

To showcase the extensibility of our framework for more challenging tasks, lacking easily available ground truths, we have implemented a metric-driven controller on an open-ended discovery task which is detailed in this section.

## 3.1 Task setup

We designed the open-ended evolution task to test if the Hypothesis Game can grow and refine a plausible biological hypothesis given very sparse prior knowledge. Unlike reconstruction or corruption tasks, which are anchored to a reference pathway, here the system must iteratively extend a minimal seed hypothesis into a richer mechanistic explanation.

The task focuses on a foundational question in drug discovery: what is the mechanism that links a gene to disease biology? We seed this task with gene-disease pairs having a strong genetic association predicted between a gene and a disease. There is little prior knowledge in biomedical literature about a potential mechanism between a gene and a disease otherwise.

We followed genetics-first approach, since it was demonstrated previously (Minikel et al., 2024), that a strong genetic association between gene targets and diseases correlated with their success in clinical trials.

We used the Open Targets (Buniello et al., 2025) database to attribute the genetic associations and literature precedence. Gene-disease pairs with a genetic association are defined by population genetic studies in the Open Targets database. Gene-disease pairs with low literature evidence are defined by the gene-disease pair AND any related diseases (defined by ontological similarity $> 0.7$ as per (Minikel et al., 2024)) having a literature score of $< 0.3$ in the 'europepmc' text mining subset of the Open Targets database.

We selected Amyotrophic Lateral Sclerosis (ALS) as a test case given its multiple strongly associated genes and the limited understanding of how these genes contribute mechanistically to

disease progression. We run predictions for three genes: *LINGO2, CFAP410* and *TYW3*. As this is an open-domain reasoning task, we updated the retrieve evidence move to use OpenAI's websearch tool instead of querying a corpus for all versions of the game shown in this section. The other moves are unchanged.

The instructions for this task were:

Role: Open-ended task instructions, Model: GPT4o

```
Your task is to develop a mechanistic scientific hypothesis.

Present the hypothesis as a causal sequence of interactions between molecular
↪   biomedical entities.

base_hypothesis: {{ gene }} impacts {{ disease }}
```

## 3.2 Controller

To support metric-driven exploration, we modified the game setup to not just iteratively update the hypothesis, but also to be able to explore various trajectories for the same hypothesis. In this version, the game trajectories are represented as a tree, where each node is a hypothesis, and the edges between the nodes are the moves used to update the hypothesis. This version of the game is a variant of Monte Carlo Tree Search (Browne et al., 2012) (MCTS).

The following steps are executed during each turn of the game:

- **Selection**: A new node (hypothesis) is selected for expansion, this is done by applying the UCT formula until we reach a node that is not fully expanded (In this experiment it has less than 2 children).

- **Expansion**: The selected node's hypothesis is passed to the Game Master that picks and executes one of the moves.

- **Backpropagation**: The resulting hypothesis is added as a children to the node that was chosen during the selection phase. The hypothesis is evaluated based on the scoring function and all the parent nodes get updated with the resulting score and visitation count.

The advantage of this setup is that we can explore alternative moves leading to different hypotheses during each game run. We used the Upper Confidence Bound for Trees algorithm (UCT) (Kocsis et al., 2006) for selecting nodes for expansion which helps in balancing exploration (focusing on less explored hypotheses) and exploitation (focusing on the most promising hypotheses). The formula for UCT is shown in equation 1. Due to the large amount of possible moves we can execute on each node (four move types, but lots of possible instructions that may be passed for them), we limited each node to only have up to 2 children to bias the tree construction to build deeper trees as we are interested more developed hypotheses.

$$UCT = \frac{Q_i}{N_i} + C\sqrt{\frac{\ln N_p}{N_i}} \tag{1}$$

where

- $Q_i$ is the total reward accumulated at child $i$.

- $N_i$ is the visit count of child $i$.

- $N_p$ is the visit count of the parent

- $C$ is the exploration constant ($\sqrt{2}$ in our experiments as commonly used in MCTS)

We developed a simple scoring function for the biological plausibility of a hypothesis. A gene-level representation was derived from a generic gene-gene interaction network (Szklarczyk et al., 2023). In short, for each gene in the interaction network, heat was diffused from the focal node for three different time steps to represent local, intermediate and global interactions. The resulting vectors were concatenated to an overall gene-level representation.

The scoring function used to guide the game in tree search experiments calculates the mean cosine similarity between the genes in the hypothesis, multiplied by the number of genes present in the hypothesis. This score encourages the game to elaborate hypotheses with many genes that are functionally related - given some curated biomedical knowledge.

## 3.3   Results

For all three selected genes (*LINGO2*, *CFAP410* and *TYW3*), the Hypothesis Game with MCTS produced hypotheses that were judged plausible by qualitative expert review ($N = 3$). Evaluating such open-ended hypotheses is inherently challenging. Beyond plausibility there is no straightforward ground truth, the space for possible mechanisms is vast and sparsely validated. More extensive experiments and more systematic evaluation strategies will be necessary to more robustly quantify hypothesis quality in future work.

As an illustrative example, we show hypotheses generated for one of the ALS-associated genes (*LINGO2*). The Hypothesis Game performs around 7 moves until it reached the shown hypothesis which used the same budget as the Hypothesis Game-MCTS (due to branching, it did not reach the same depth). We also show a hypothesis that was generated when 50 moves were allowed for MCTS version. The increased move budget has resulted in a noticeably more granular hypothesis generated by Hypothesis Game-MCTS.

---

**Base hypothesis: LINGO2 impacts amyotrophic lateral sclerosis (Hypothesis Game)**

```
LINGO2 impacts amyotrophic lateral sclerosis by modulating axonal degeneration and
↪  regeneration pathways, potentially through interactions with axonal growth
↪  inhibitors such as Nogo, MAG, or OMgp. LINGO2 is expressed in central nervous
↪  system neurons in healthy tissues, but its specific expression in motor
↪  neurons-particularly in ALS-affected tissues-remains unconfirmed, highlighting a
↪  critical gap in understanding its role in motor neuron pathology. Single-cell
↪  RNA-seq data from the Human Protein Atlas indicate that LINGO2 is enriched in
↪  excitatory and inhibitory neurons, supporting its neuronal relevance, though
↪  direct evidence of expression in motor neurons is still lacking. Recent in situ
↪  hybridization studies in ALS models have not reported LINGO2 expression in motor
↪  neurons, underscoring the need for targeted investigation of its spatial
↪  expression in ALS-affected tissues. Notably, LINGO2 shows robust neuron-specific
↪  expression in motor-relevant regions such as the caudate nucleus of the basal
↪  ganglia, suggesting a potential role in motor circuitry even in the absence of
↪  direct motor neuron expression data. Furthermore, single-cell RNA-seq clustering
↪  places LINGO2 within the 'Neurons - Neuronal signaling' group with high
↪  specificity (Tau $\approx 0.92), reinforcing its functional association with
↪  neuronal signaling pathways in healthy human brain tissue.
```

## Base hypothesis: LINGO2 impacts amyotrophic lateral sclerosis (MCTS)

```
LINGO2 impacts amyotrophic lateral sclerosis by modulating axonal degeneration and
↪   regeneration p
athways, potentially through interactions with axonal growth inhibitors such as Nogo,
↪   MAG, or OMgp. Although direct evidence is lacking, LINGO2 has been shown to
↪   physically interact with other membrane proteins such as TFF3 and EGFR via
↪   co-immunoprecipitation and proximity ligation assays, supporting the plausibility
↪   of similar interactions with Nogo, MAG, or OMgp. Additionally, LINGO2 is
↪   upregulated in injured human neurons in Alzheimer's disease models, where its
↪   knockdown rescues neurite outgrowth and reverses transcriptomic signatures of
↪   synaptic dysfunction and apoptosis, suggesting a broader role in
↪   neurodegenerative processes that may extend to ALS. Proximity ligation assays
↪   have successfully detected LINGO2 interactions in situ, establishing a feasible
↪   method to investigate potential LINGO2-MAG interactions in fixed tissues or cells.
↪   While no direct evidence currently links LINGO2 to OMgp in ALS, the
↪   well-characterized interaction between OMgp and LINGO1 in axonal growth inhibition
↪   via the NgR1 complex suggests a plausible mechanistic parallel for LINGO2, given
↪   its structural homology and neuronal expression.
```

> **Base hypothesis: LINGO2 impacts amyotrophic lateral sclerosis (MCTS 2x budget)**
>
> ```
> LINGO2 impacts amyotrophic lateral sclerosis by modulating axonal degeneration and
> ↪  regeneration pathways, potentially through interactions with axonal growth
> ↪  inhibitors such as Nogo, MAG, or OMgp. LINGO2 is expressed in healthy
> ↪  motor-relevant neurons, including those in the cortex, spinal cord, and dorsal
> ↪  root ganglia, suggesting a neuron-specific role in central nervous system
> ↪  function. Although no direct evidence currently demonstrates a physical
> ↪  interaction between LINGO2 and Nogo, MAG, or OMgp, the structural similarity of
> ↪  LINGO2 to LINGO1-which forms complexes with Nogo receptor components-along with
> ↪  the availability of co-immunoprecipitation-compatible LINGO2 antibodies, supports
> ↪  the feasibility of experimentally testing such interactions. LINGO2 may also
> ↪  modulate axonal degeneration and regeneration by influencing the RhoA/ROCK
> ↪  signaling cascade, a key intracellular pathway involved in growth cone collapse
> ↪  and axonal retraction. Recent evidence from human iPSC-derived neuron models and
> ↪  rodent expression studies suggests that LINGO2, like its homolog LINGO1, may
> ↪  negatively regulate neurite outgrowth and synaptic function via
> ↪  RhoA/ERK-associated signaling, supporting its potential role as an upstream
> ↪  modulator of RhoA/ROCK pathway activity in neuronal contexts. Furthermore, LINGO2
> ↪  has been experimentally shown to physically interact with other membrane proteins
> ↪  such as EGFR via co-immunoprecipitation, validating the use of this technique to
> ↪  probe potential interactions with Nogo receptor components. In human iPSC-derived
> ↪  neurons carrying the APP^V717I mutation, LINGO2 knockdown rescues neurite
> ↪  outgrowth deficits and reverses activation of ERK/MAPK, p53, and apoptotic
> ↪  signaling pathways, suggesting a functional role for LINGO2 in modulating
> ↪  neuronal survival and growth signaling. Although no direct evidence currently
> ↪  links LINGO2 to RhoA/ROCK signaling in ALS models, elevated ROCK activity has
> ↪  been observed in SOD1-G93A ALS mice and human ALS tissues, and pharmacological
> ↪  ROCK inhibition improves motor function and axonal regeneration, supporting the
> ↪  relevance of this pathway in ALS pathophysiology. Co-immunoprecipitation studies
> ↪  have demonstrated that LINGO2 can form stable protein complexes with
> ↪  membrane-associated partners such as EGFR and BK channels, confirming its capacity
> ↪  to engage in physical interactions with transmembrane proteins and supporting the
> ↪  plausibility of similar interactions with Nogo receptor components.
> ```

Beyond looking at the generated hypotheses, the game trajectories produced by the tree-based controller highlight many cases where trying alternative moves from a state can lead to more interesting hypotheses. Since the linear version of Hypothesis Game has no mechanism to go back to previous hypothesis states it often stops improving the hypothesis after some time. The advantage with the MCTS-based controller is that it allows going back to previous states and create new trajectories from there often leading to improved hypotheses. With MCTS we can also increase the move budget to explore more hypothesis and greater depths (more subsequent moves applied to them).

These preliminary results have shown that our framework can be extended with MCTS to produce hypotheses better aligned with our defined scoring functions. The experts who reviewed the generated hypotheses, found them plausible and interesting. As future work, we would like to further improve the controller and perform larger-scale benchmarking.