# OpenReview forum: "Tiny Moves: Game-based Hypothesis Refinement"
_ICLR.cc/2026/Conference — Submitted to ICLR 2026_

### Official Review · Reviewer_hak6 · 2025-10-26

**Soundness:** 3
**Presentation:** 3
**Contribution:** 2
**Rating:** 4
**Confidence:** 3

**Summary:**

This paper introduces Hypothesis Game, a symbolic, game‑based framework in which LLM agents refine a shared hypothesis via a fixed grammar of small “tiny moves” (actions). Hypotheses are represented as sets of fragments, and the experiments primarily use structured text. The implemented move set comprises prune, expand, expand_with_corpus (retrieve + integrate), and debate, orchestrated by a central LLM controller called the Game Master. Modes guide move selection through prompt instructions in the prototype, and a scoring formalism is defined but not used to drive control. Two game variants are specified: Simple Hypothesis Refinement (whole‑state edits) and Localized Hypothesis Refinement (fragment‑level edits with consistency enforcement), each given as an algorithm.

The evaluation focuses on pathway‑level reasoning from the Reactome dataset through two tasks: 1. reconstruction from partial cues and 2. corruption recovery. The study samples 100 pathways for reconstruction and 20 for corruption, running 820 and 2880 experiments, respectively. The authors compare the proposed one with baselines: Zero‑Shot, Chain‑of‑Thought, and ReAct; metrics combine entity‑level precision/recall/F1 via Gilda mapping and an LLM‑as‑judge measure of reaction‑level “Detailed Recall.” In reconstruction, all methods struggle; the Hypothesis Game is comparable to ReAct, with text noting ReAct’s slightly higher F1.
In corruption recovery, the Hypothesis Game could remove more errors and achieves the highest precision and F1 while maintaining recall.

**Strengths:**

- The writing is very clear and very easy to follow. I like the way the authors illustrate the key idea and its association with human science development.
- The high-level idea makes sense. This work formalizes hypothesis refinement as a compositional reasoning game with a reusable grammar of moves, enabling transparent trajectories and controllable reasoning styles.
- The approach demonstrates relatively good performance in corruption recovery, combining higher error removal with the high precision and F1 while maintaining recall.
- The paper releases curated datasets on Hugging Face for reproducibility.
- Related works are generally sufficient there.

**Weaknesses:**

- The high-level idea of employing LLM for action planning given a set of operators is not very novel.  I am not very sure about the significance of the novelty/technical contribution of this paper - currently im not very positive.
- Consider adding more comparative agent frameworks as baselines. Also, the experiment scope is constrained to Reactome human pathways and an open‑access biomedical corpus, which is a bit limited.
- The performance is not "that" good on such a scope of experiment. In the reconstruction task, ReAct actually still slightly outperforms the Hypothesis Game in F1, and all methods show low precision and recall.
- Is an LLM‑as‑judge for reaction‑level correctness reliable?
- The method needs \(k_{\max}\) and termination criteria, but the experiments vary only move availability and corpus access. The move‑selection prompt instructs “run at least 20 rounds,” which interacts with traceability and cost but is not ablated.
- Minor: Fig.1 in the appendix is not clear. In the appendix: "The moves used in this section correspond to the moves presented in table ??.", please fix this..

**Questions:**

Besides the question raised above.

- The reaction‑level metric uses an LLM‑as‑judge; can you report any calibration or agreement checks to assess the judge’s reliability?

**Details Of Ethics Concerns:**

/

---

> ### Author Response · Authors · 2025-11-20
> **grouped responses to the reviewer's comments**
>
> Thank you for the helpful feedback. Below we address your points.
> 1. [**General novelty and contribution**]: We understand the concern. Our main contribution is not the use of operators alone, but the formulation of structured scientific hypothesis refinement as a game with a concrete set of moves that can be both agnostic and domain-grounded. We also outline how the reasoning process could be guided and evaluated through a utility function and introduce two benchmark tasks tailored to biological discovery. To our knowledge, this is the first work to formalize and evaluate multi-step operator-driven refinement for mechanistic biological hypotheses, which involve uncertainty, context-dependence, and latent structure distinct from standard planning settings. Our experiments aim to demonstrate the value of this formulation, rather than propose a new planning algorithm.
> 2. [**Baselines**]: We agree more comparisons are useful, but to the best of our knowledge there are no established benchmarks that evaluate multi-step mechanistic reasoning of the kind studied here. We selected CoT-based variants because in our view they provide the fairest available comparison. Related systems such as BioMaze or Co-Scientist tackle biology-inspired reasoning but either rely on specialized tools, focus on not directly comparable tasks, or lack accessible implementations, which limits direct comparison. We would welcome suggestions of specific frameworks the reviewer considers appropriate for this domain.
> 3. [**Reactome Corpus**]: We focused on Reactome pathways and open-access biomedical papers to ensure the setup is fully reproducible and to avoid licensing constraints associated with proprietary corpora.
> 4. [**Performance**]: We agree that overall performance in the reconstruction task is low across the board. This reflects the difficulty of reconstructing full mechanistic pathways from partial cues. This task requires deep domain expertise and multiple reviews by expert curators when attempted manually. As we discussed in the paper, three factors could contribute to the low scores: 1) limited information in the partial inputs, 2) heterogeneity in pathway curation, and 3) the relatively abstract level of detail available in the open-access corpus. Even though some pathways are established and at least partially represented in LLM pretraining data, this does not translate into high recall for any method. This highlights the challenge of recovering precise mechanistic structure from partial cues, as well as difficulty in establishing realistic reasoning benchmarks for biology in general.
> While ReAct’s achieves slightly higher F1 on reconstruction, this difference is not statistically significant (updated Figure 3). Our goal in this task was to evaluate if the tiny-move refinement approach is viable in such a challenging setting, rather than to claim superiority. Importantly, in the corruption-recovery task, where the input is richer, the our approach achieves the best precision and F1, suggesting that it provides value when sufficient domain information is available.
>
> 5. [**LLM-as-judge**]: We agree that a formal comparison of expert annotators with the LLM-as-judge evaluation is necessary. We update the manuscript to include a comparison of 2 domain-expert annotators against the LLM-as-judge for the reconstruction and corruption tasks (Suppl. A, Section 3.7.3). All annotations were fully blinded.
> For the corruption task, Krippendorff’s α was ≥ 0.9 for all human-human and human-LLM comparisons (Suppl. A, Table 2), indicating that the LLM judge closely tracks expert judgement.
> For the reconstruction task, Krippendorff’s α was substantially lower for the human-LLM inter-annotator agreement (Suppl. A Table 3), specifically around the presence/absence of participating biomedical entities. A qualitative assessment of the disagreements showed the LLM-as-judge was stricter and considered specific family members to be incorrect entities, whereas human annotators considered specific exemplars as representative of general, well-established signalling pathways. Despite the misalignment between human and LLM annotations, the trend is qualitatively the same between the detailed reaction-level recall (LLM-as-judge) and the entity recall (NER).
>
> 6. [**Termination criteria**]: We found that in most cases games terminate before they reach the maximum number of rounds (25). Importantly, neither the agents nor the Game Master have access to the current round count, and k_{\max} serves only as a safety upper bound, not as a control parameter. The prompt simply encourages the Game Master to avoid stopping prematurely with an underdeveloped hypothesis. Because termination is determined by the controller’s Diagnose component rather than by a fixed budget, varying k_{\max} or the nominal round guidance does not influence traceability, cost or comparative results in our experiments.
>
> 7. [**Formatting**]: we have now updated the reference.

---

### Official Review · Reviewer_3i5p · 2025-10-30

**Soundness:** 3
**Presentation:** 3
**Contribution:** 3
**Rating:** 6
**Confidence:** 2

**Summary:**

The paper introduces The Hypothesis Game, a symbolic, game-based framework for iterative hypothesis refinement using a small reasoning grammar ("tiny moves") applied to a shared hypothesis state \(H_t\). The formalism includes game modes (via \(\pi_M\)) and an optional scoring vector \(S(H_t)\), with two algorithmic variants for whole-state and localized edits. Experiments target pathway-level reasoning on Reactome: (i) reconstruction from partial cues and (ii) recovery from corrupted hypotheses. Using entity-level metrics (via Gilda) and an LLM-as-judge metric for reaction fidelity, the method matches strong prompting baselines on reconstruction and achieves higher precision and F1 on corruption, across 820 reconstruction and 2,880 corruption trials.

**Strengths:**

- The paper formalizes hypothesis refinement as iterated transformations under a move budget, providing a crisp, compositional semantics. This clarity aids reuse and analysis.

- A small set of moves (prune, expand, retrieve, debate) is specified and mapped to agent responsibilities. This fosters generality while remaining practical. The grammar supports composition and repeated application, encouraging modular reasoning and stepwise traceability. Conceptual visualization (Fig. 1) improves readability for non-experts and illustrates extensibility to graph settings.

- The "Game Master" controller and specialized agents are described, with moves and diagnostics summarized in Table 1. Modes are approximated via prompt conditioning (Sec. 3), clarifying how theory is realized in practice.

- Evaluation tasks are well-motivated. Two tasks (reconstruction and corruption recovery) are well aligned with incremental hypothesis evolution (Sec. 4). Entity fidelity uses standardized mapping (Gilda) and reaction fidelity uses an LLM-as-judge checking inputs/outputs/directionality/interaction type. This provides complementary views. Dataset size and split are substantial for a pilot.

- The method achieves the highest precision and F1 in corruption recovery across error types and rates, showing the value of targeted edits. On reconstruction, performance is close to ReAct and better than Zero-Shot/CoT on precision, indicating disciplined expansion helps avoid spurious content.

**Weaknesses:**

- Scope of baselines and diagnostics is narrow. Baselines are limited to prompting (Zero-Shot, Chain-of-Thought, ReAct), omitting other structured controllers or graph-based planners, which constrains external validity. No ablation on the move grammar (e.g., removing debate or retrieve) or on the move budget to isolate which components drive gains.

- Formal elements are not fully operationalized. Modes \(\pi_M\) are implemented via prompt text rather than as an explicit policy; scoring \(S(H_t)\) and \(U(H_t)\) are defined but not used to drive control. Optimization is handled by the controller’s "Diagnose" component without metric-driven selection, weakening the empirical link between the formalism and gains. No experiments vary \(\beta\) or demonstrate policy learning from scores, leaving the control levers untested.

- LLM-as-judge reliability and human oversight are under-specified. Reaction-level fidelity relies on an LLM judge*with no reported agreement against human adjudication or across multiple judges. Although two experts review corruption generation, inter-annotator agreement is not reported.

- Potential leakage/contamination risks are not controlled. The authors note that some pathways are "relatively well known", which may favor models with prior knowledge or retrieval, but no time-based or contamination controls are described. Retrieval-based moves can draw on external corpora, yet there is no ablation on retrieval sources or temporal cutoffs. No assessment is given for how performance changes when restricting to knowledge post-dates or unseen pathways.

**Questions:**

See Weakness

---

> ### Author Response · Authors · 2025-11-20
> **grouped response to reviewer's comments**
>
> Thank you for the helpful feedback. Below we address your points.
> 1. [**Baselines**]: We agree that broader comparisons would be valuable. However, to our knowledge there are no established multi-step reasoning baselines for realistic biological hypothesis refinement that are sufficiently decoupled from specialized tools or domain-specific corpora. Existing systems such as BioMaze or Co-Scientist target different problem formulations and do not provide a clean, directly comparable baseline for operator-based refinement. To ensure a fair comparison, we selected Zero-Shot, CoT, and ReAct as the strongest general-purpose methods aligned with our setting, allowing us to isolate the contribution of the tiny-move grammar without introducing domain-specific confounders. Multi-step biological reasoning benchmarks are still emerging, and broadly applicable baselines remain limited; we would welcome suggestions of specific frameworks the reviewer considers suitable.
>
> 2. [**Ablations**]: We include move-type ablations for the reconstruction task (20 Reactome pathways; Supp. B, Table 1,  Sec 1.3). We did not extend these to the corruption task, as its multiple interacting error types and rates make controlled removal of specific moves difficult to interpret. We agree broader ablations would be valuable and plan to explore them in follow-up work as we develop more comprehensive evaluation strategies for complex biological hypotheses.
>
> 3. [**Framework vs implementation**]: We agree that the current implementation does not instantiate all components of the formalism; this was intentional. Outlining the broader design helps clarify how the pieces fit together and provides a foundation for future work, though we recognize this may be confusing and welcome suggestions on improving clarity. To address this, we added a paragraph to Section 2 explicitly distinguishing the minimal instantiated implementation from the full formalism.
> Our goal was to introduce the framework and evaluate the reasoning-move grammar under the simplest viable controller. Modes are defined via natural-language instructions, and we intentionally did not use the scoring function, as it was unnecessary for the two controlled tasks. We acknowledge that the prompt-based controller offers limited control, and more sophisticated controllers will be needed for open-ended discovery. As an illustration, we implemented a lightweight Monte Carlo tree search variant (Supp. B, Section 3). In an open-ended task on ALS-associated genes with limited prior knowledge, this MCTS-based variant produced plausible hypotheses for three genes based on expert judgement; systematic quantitative comparison is left for future work.
>
> 4. [**LLM as judge**]: We agree that our original description lacked detail on the corruption bank and LLM-as-judge calibration. The corruption bank was not created through independent double-annotation with a reportable IAA; instead, LLM-generated candidate corruptions were iteratively curated by two senior experts, who refined prompts and manually corrected statements over multiple rounds until each example matched the intended error type and difficulty (updated Section 2.1). To assess LLM-as-judge reliability, we added a post-hoc calibration. The same two experts independently evaluated model outputs for both tasks, and we computed human–human and human–LLM agreement. For corruption, experts labelled a stratified sample of 20 model-corrected pathways, blinded to all conditions, and we computed Krippendorff’s α on the rater-by-item matrix. All human–human and human–LLM comparisons yielded α ≥ 0.9 (Suppl. A, Sec 3.8, Table 2), indicating close alignment with expert judgements. For reconstruction, human–LLM α was substantially lower (Suppl. A, Sec 3.8, Table 3), primarily due to stricter entity matching by the LLM (e.g., treating specific family members as incorrect where experts viewed them as valid representatives of canonical pathways). Despite this mismatch, the qualitative trend remains consistent between reaction-level recall (LLM-as-judge) and entity-level recall (NER).
>
> 5. [**Leakage**]: Given that the task is to reconstruct or repair curated pathways from their own cited evidence, imposing an artificial temporal cutoff or substituting unrelated corpora would distort the task rather than provide a meaningful contamination control. All curated Reactome pathways in our dataset were built from biomedical papers published before the GPT-family pretraining cut-off, so there are no “unseen pathways” for which a time-based control would be informative. We also report an ablation without the corpus (Suppl. B Section 1.3.2), showing that retrieval improves performance. Each Reactome pathway is linked to the primary literature used during its curation, and our reconstruction task mirrors this process, except that retrieval is restricted to pathway-specific open-access citations rather than the full biomedical literature.

---

### Official Review · Reviewer_1dmE · 2025-10-31

**Soundness:** 3
**Presentation:** 3
**Contribution:** 3
**Rating:** 6
**Confidence:** 3

**Summary:**

The paper introduces "The Hypothesis Game", a symbolic, game-based framework for hypothesis refinement in scientific discovery. The method formalizes reasoning as a set of discrete operations ("tiny moves") such as prune, expand, retrieve, and debate, applied iteratively to structured hypotheses. Evaluation on pathway-level reasoning tasks derived from Reactome shows that the approach performs comparably to strong prompting baselines in hypothesis reconstruction and outperforms them in error correction.

**Strengths:**

1) The paper provides a principled and interpretable alternative to free-form LLM reasoning.
2) The reasoning grammar is well motivated and modular, allowing transparent control of reasoning styles and easy extensibility.
3) The experimental setup is carefully designed, including both reconstruction and corruption tasks on Reactome pathways.

**Weaknesses:**

In my opinion, the main weakness of the paper is that the scoring formalism proposed in Section 2.4 remains theoretical and untested.
Other than that:
1) The evaluation lacks any assessment of statistical significance across multiple runs. This limits confidence in the reported performance differences.
2) Biological validation of the refined hypotheses is absent. While metrics are informative, there is no verification that reconstructed or corrected pathways remain biologically plausible or mechanistically consistent.

**Questions:**

I would ask the authors to address the weaknesses highlighted above. Particularly, the authors should either implement one scoring-driven variant to illustrate how metrics could guide reasoning trajectories, or provide more explanations on why  scoring-driven variants are left out from the current work.
Moreover:
1) Please assess the statistical significance of the reported performance differences between The Hypothesis Game and the baselines.
2) Include at least a few biologically grounded evaluation, such as checking whether (a randomly chosen subset of) reconstructed pathways preserve known functional or causal relationships.

---

> ### Author Response · Authors · 2025-11-20
> **grouped response to reviewer's comments**
>
> 1. [**Evaluation**]: We appreciate the reviewer's point regarding statistical significance. We now include significance estimates comparing our method to the baselines, see updated Figure 3 and figure legend.
> Regarding the significance across multiple runs, due to the computational cost of full multi-seed evaluation over 3,700+ trials, we were not able to rerun all experiments under multiple random seeds within the rebuttal window. However, our experimental design already includes a built-in robustness check. The corruption task evaluates performance across three types of errors, two levels of difficulty and varying error rate. Across all variants of the corruption experiment, the performance trends are highly consistent (see Supplementary B, Figures 2 and 3). In our view, this stratification provides evidence that the observed performance differences are robust and not driven by a specific corruption setting. We have added 95% confidence intervals for all results to demonstrate this.
> We agree that additional multi-seed replications would further strengthen the reconstruction analysis. Within the rebuttal window, however, generating full multi-seed runs was not feasible..Our current reconstruction results already average over a large number of independent pathway instances (820 trials across 100 pathways), where pathway-specific difficulty provides the main source of variability. We therefore treat pathways as the unit of replication and report confidence intervals obtained by bootstrapping over pathways (Figure 3); under this resampling, the relative ordering of methods is unchanged. We plan to include full multi-seed replications in future work.
> 2. [**Scores**]: We intentionally implemented the simplest functional version of the game, including an LLM-based controller, to isolate the contribution of the move grammar itself. We did not omit the scoring-driven controller by oversight. Based on our preliminary experiments, stable optimization requires a reliable reward signal, which we plan to derive from wet-lab experiments, as we came to discover metrics derived from text-based hypotheses alone can be too noisy. We have clarified this limitation in the revised scoring section. To address the reviewer’s suggestion and to illustrate how the formalism can be extended beyond the minimal implementation, we also added a paragraph in Section 2 (Framework) distinguishing the evaluated minimal setup from the broader design space. In addition, we implemented a simple scoring-guided variant using Monte Carlo tree search, driven by a gene-gene similarity score (Supplementary B, Section 3). In this open-ended discovery task, the goal was to propose mechanisms of action for ALS (amyotrophic lateral sclerosis)-associated genes with limited prior mechanistic knowledge. Running the tree-based tiny-moves version on three such genes produced hypotheses judged plausible by experts. A thorough quantitative comparison with baselines is left for future work.
> 3. [**Biological validation**]: Our evaluation is grounded in known biology by design. Every reconstructed or corrected pathway is compared directly against its curated Reactome reference, which encodes experimentally established functional and causal relationships. In the corruption task, all errors are synthetically introduced into a known ground-truth pathway, so we can measure if the game restores the correct biological interactions. In the reconstruction task, generated pathways are evaluated against the same curated Reactome ground truth, and our entity- and reaction-level metrics capture recovery of key functions and causal directions. We agree that biological validation is essential. Our focus in this paper was to establish the reasoning framework and evaluate its behavior in controlled settings, where ground truths are at least partially available. Performing experimental validation, such as testing if an output hypothesis is mechanistically sound and biologically plausible, is unfortunately not feasible within the rebuttal timeframe as it requires considerable wet lab work. We are actively developing a complementary framework for experimental validation of mechanistic hypotheses. The goal is to identify hypothesis fragments with the highest uncertainty and design targeted assays (e.g., protein-protein binding or perturbation experiments) in the appropriate biological context (cell type, tissue) to test them directly. Incorporating such experimental signals, even as sparse high-value rewards is a key direction of our ongoing work.

---

### Official Review · Reviewer_9Mpq · 2025-10-31

**Soundness:** 2
**Presentation:** 2
**Contribution:** 2
**Rating:** 4
**Confidence:** 3

**Summary:**

The paper introduces The Hypothesis Game, a game-based framework in which LLM agents refine scientific hypotheses through small, structured reasoning steps defined by four operations: prune, expand, retrieve, and debate. It models hypothesis refinement as an iterative process guided by a fixed reasoning grammar. The approach is evaluated on biological pathway reasoning tasks from Reactome, including reconstruction and corruption recovery. In experiments, it performs comparably to strong prompting baselines on reconstruction and achieves higher precision and F1 scores on corruption recovery.

**Strengths:**

(1) The paper proposes a novel framing of scientific reasoning as a game with explicit, interpretable moves, offering an original perspective on hypothesis refinement.

(2)The paper is well organized and clearly written, making a complex concept easy to follow.

**Weaknesses:**

* The scope is limited to biological pathways; it is unclear how the framework performs in other scientific domains.
* The controller is still prompt-based, not a learned or metric-driven system, which makes the implementation less autonomous.
* The move set is small (only four operations), which may limit creativity and open-ended discovery.
* The evaluation relies partly on LLM judgments and does not include human or symbolic verification.

**Questions:**

* The scoring function $U(H_t)$ is defined but unused. It would be useful to clarify what prevented its use and how explicit scoring might affect the controller’s reasoning.
* The paper often refers to 'game', but the implementation appears to be prompt-based rather than a true multi-agent or turn-based game. Clarification on whether it is primarily a conceptual framing or includes actual game dynamics such as turns, scoring, or agent interaction would strengthen understanding.
* How consistent is move selection across runs? An evaluation of the stability of this prompt-based control would be informative.
* The framework employs four core operations (prune, expand, retrieve, debate). Further explanation of why these specific operations were chosen and whether adding or removing move types affects performance would clarify the design space.
* The paper mentions that pathway descriptions were rephrased "to avoid memorization," but the procedure is not detailed. It is unclear who performed this rephrasing (humans or LLMs) and how semantic accuracy was verified would improve clarity and reproducibility.
* How was semantic equivalence verified after rephrasing, and could rewording introduce ambiguity or loss of information?
* Did the two experts agree with each other when reviewing the corruptions?
* Did those same experts also check whether the model’s corrected hypotheses were actually right after the experiments?
* Figure 2 -- Qualitative Example: The example compares incremental vs. single-step edits. How was "minor additional change" quantified? Is it possible to use token-level or entity-level edit distances?
* The framework aims to support hypothesis discovery, but its refinement process prioritizes consistency with known data. Could there be a discussion on whether iterative editing might reduce the novelty of initially creative hypotheses, effectively converging toward familiar knowledge rather than exploring new ideas?

---

> ### Author Response · Authors · 2025-11-20
> **summary response to reviewer's comments**
>
> 1. [**Scope**]:  Thank you for raising this limitation. We focus on biological pathways because they offer a controlled yet complex setting with high context-dependence and uncertainty, and there are no established benchmarks for general scientific reasoning in biology. While reductionistic, these tasks allow systematic evaluation. The underlying game formalism and move grammar are domain-agnostic and can be extended to other scientific problems; we plan to explore this in future work.
>
> 2. [**Controller**]: Our aim is to introduce the framework for structured hypothesis refinement as an operator-based game. The controller is only one component and can be instantiated in many ways; here we use a prompt-based LLM controller as a minimal implementation for evaluating the move grammar. Learned or metric-driven controllers (e.g., RL-based) is feasible within the formalism. The scoring function was included to illustrate how explicit utilities could be integrated, but we did not use it because reliable signals for biological hypotheses likely require experimentally grounded or expert-curated feedback. We revised Section 2.4 to clarify this distinction between the conceptual framework and the minimal prototype.
> Although minimal, the implementation is turn-based and multi-agent: the Game Master selects moves and delegates to specialised agents operating on a shared hypothesis state. “Game” is primarily a conceptual framing, and the prototype instantiates only the core elements needed for evaluation. We added text in Section 2 distinguishing the minimal implementation from the full formalism. To illustrate extensibility, we implemented a Monte Carlo tree search variant guided by a gene-gene similarity score (Supplementary B, Section 3).
>
> 3. [**Moves**]: We began with a small, domain-agnostic move set to show that even minimal operators can support multi-step refinement. Additional moves can be added for richer hypothesis representations or more complex tasks. The four core moves correspond to fundamental interactions: expand (grow), prune (shrink/reassess), retrieve (ground in evidence), and debate (challenge/critique). Prune and expand are necessary counterparts, and retrieve ensures revisions remain evidence-linked.
> Moves also combine compositionally: order matters (e.g., prune→expand vs. expand→prune), and each targets different hypothesis components, yielding a larger effective action space than the raw move count. The Game Master’s instructions further specify which components to modify, providing additional contextual control. Overall, this move set is a minimal, interpretable starting point. Ablations for the reconstruction task appear in Supplementary B, Section 1.3.
>
> 4. [**LLM-as-judge**]: Rephrased pathway statements were generated with an LLM and checked by a domain expert to ensure that biological meaning and directionality were preserved; ambiguous or incorrect rewrites were corrected or removed. These details are now in Section 4.1, and all original and rephrased titles are available in the HuggingFace dataset.
> The corruption bank was not created via independent annotations with measurable agreement but via an iterative collaborative process: LLM-generated corruptions were jointly reviewed and corrected across multiple rounds by two experts until each matched the intended error type and difficulty. We clarified this in Section 4.1 and Supp. A, Section 2.1.
> We also added a post-hoc calibration: two experts independently annotated a stratified sample of 20 model-corrected pathways to judge whether the original corruption persisted. Annotations were fully blinded, and Krippendorff’s α ≥ 0.9 for all human–human and human–LLM comparisons (Supp. A, Table 2), indicating close alignment between the LLM judge and expert assessments.
>
> 5. [**Fig 2**]: In Fig. 2, “minor additional change” corresponds to quantitative metrics shown in Supplementary B Fig. 4: (1) gene-level entity additions/removals using Gilda mapping and (2) word-level normalized Levenshtein distance to the reference description. We revised the main text and caption to make this explicit.
>
> 6. [**Discovery**]: We agree that balancing consistency and novelty is central to discovery. In this paper we validate the framework in a controlled, ground-truth setting, so the controller is tuned toward biologically plausible refinements rather than open-ended exploration. However, the formalism is designed to support more divergent reasoning: adjusting the controller, move distribution, or expansion operators can steer the system toward novelty. Fully unconstrained exploration is hard to validate, and structured extensions of known mechanisms are often more actionable. Importantly, novelty often arises through recombining existing fragments into new mechanistic configurations; the Hypothesis Game supports such compositional novelty, so iterative refinement need not suppress creativity.

---

### Author Response · Authors · 2025-12-01
**Summary**

We sincerely thank the reviewers for their thoughtful and constructive feedback, which has helped to improve and strengthen our manuscript. Below we summarize the key contributions of our paper and key additions from the rebuttal.

## Key contributions
- **Game-based formalism for hypothesis refinement:**
We present _**HypothesisGame**_, which models scientific hypothesis refinement as a sequence of small, explicit operators on a shared hypothesis state. A compact move grammar (prune, expand, retrieve, debate) supports both whole-state and localized edits. This turns free-form LLM reasoning into transparent and composable transformations.

- **New benchmark setting for mechanistic biological reasoning:**
 We instantiate this framework on biochemical pathways from Reactome with two reasoning tasks: 1) reconstruction from partial cues; 2) recovery from structured corruptions. We release all curated datasets and prompts, contributing to an early effort toward a systematic benchmark for multi-step mechanistic reasoning in biology.

- **Empirical evidence that “tiny moves” improve robustness:**
Across 3.5K+ game runs (2,880 corruption; 820 reconstruction), the Hypothesis Game outperforms all prompting baselines on the corruption recovery and performs on par with ReAct in reconstruction. On the corruption task, it achieves higher error removal, precision and F1 while maintaining recall. This shows that small, targeted edits yield cleaner and more reliable reasoning over complex mechanistic biological pathways than standard prompting.



## Key Additions from Rebuttal

- **Framework clarification:** We clarified that the LLM-based controller is intentionally minimal to isolate the effect of the move grammar, and explained why reliable reward signals are needed before deploying scoring- or RL-based controllers.

- **Extensibility illustration:** We added a tree search-based (MCTS) controller guided by gene-gene similarity for an open-ended hypothesis evolution task (Supplementary). This illustrates how the Hypothesis Game formalism can extend to metric-driven exploratory reasoning. Running the tree-based version of the game on three genes associated with ALS with unclear mechanisms produced hypotheses judged plausible by domain experts.

- **Stronger evaluation robustness:** We added confidence intervals and significance tests for all main metrics, and calibrated the LLM-as-judge against two domain experts. On the corruption task, expert agreement is high, strengthening confidence in the empirical claims findings.

---

### Meta-Review · Area_Chair_wqri · 2026-01-09

**Summary:**

This work proposed a symbolic, game-based framework for LLM agents for hypothesis refinement in scientific discovery, which is evaluated on biological pathway-level reasoning tasks. The reviewers highlighted the novel perspective of the proposed framework, careful experimental design, and clear writing. The major concerns are (1) the experimental scope is limited, specifically to biological pathways; (2) the evaluation that uses LLMs-as-Judges might not be reliable; (3) the implementation is mostly prompting-based, which limits its technical novelty, and its improvement is not very significant compared to baselines such as ReAct in some evaluations.

**Reviewer Concerns:**

The rebuttal clarifies several technical details raised in the reviews. It also provides explanations regarding concerns about the limited experimental scope and performance improvements, although these do not contradict the original reviews. Regarding evaluation reliability, the rebuttal shows that the LLM-judge and human evaluations have acceptable agreement, and that the human evaluation itself exhibits good inter-annotator agreement. However, the size of the human evaluation is limited, so these concerns are not fully addressed.

**Reviewer Scores:**

Reviewer 9Mpq is unlikely to change the score since concerns regarding the limited experimental scope are not addressed.

Reviewer 1dmE and Reviewer 3i5p are unlikely to change the score because of the remaining unaddressed concerns and the original positive scores.

Reviewer hak6 may also keep their assessment, since their concerns regarding the limited technical contribution and limited improvement are not fully addressed.

---

### Decision · Program_Chairs · 2026-01-26

Reject